# A sustained type I IFN-neutrophil-IL-18 axis drives pathology during mucosal viral infection

Tania Lebratti[1†‡], Ying Shiang Lim[1†], Adjoa Cofie[1], Prabhakar Andhey[2], Xiaoping Jiang[1], Jason Scott[1], Maria Rita Fabbrizi[1§], Ayşe Naz Ozantürk[1], Christine Pham[3], Regina Clemens[4], Maxim Artyomov[2], Mary Dinauer[5], Haina Shin[1]*

[1]Department of Medicine/Division of Infectious Diseases, Washington University School of Medicine, St Louis, United States; [2]Department of Pathology and Immunology, Washington University School of Medicine, St Louis, United States; [3]Department of Medicine/Division of Rheumatology, Washington University School of Medicine, St Louis, United States; [4]Department of Pediatrics/Division of Critical Care Medicine, Washington University School of Medicine, St Louis, United States; [5]Department of Pediatrics/Hematology and Oncology, Washington University School of Medicine, St Louis, United States

*For correspondence:
haina.shin@wustl.edu

[†]These authors contributed equally to this work

Present address: [‡]Bayer, Crop Science Division, St Louis, United States; [§]Department of Molecular and Clinical Cancer Medicine, Northwest Cancer Research Centre, University of Liverpool, Liverpool, United Kingdom

Competing interests: The authors declare that no competing interests exist.

**Abstract** Neutrophil responses against pathogens must be balanced between protection and immunopathology. Factors that determine these outcomes are not well-understood. In a mouse model of genital herpes simplex virus-2 (HSV-2) infection, which results in severe genital inflammation, antibody-mediated neutrophil depletion reduced disease. Comparative single-cell RNA-sequencing analysis of vaginal cells against a model of genital HSV-1 infection, which results in mild inflammation, demonstrated sustained expression of interferon-stimulated genes (ISGs) only after HSV-2 infection primarily within the neutrophil population. Both therapeutic blockade of IFNα/β receptor 1 (IFNAR1) and genetic deletion of IFNAR1 in neutrophils concomitantly decreased HSV-2 genital disease severity and vaginal IL-18 levels. Therapeutic neutralization of IL-18 also diminished genital inflammation, indicating an important role for this cytokine in promoting neutrophil-dependent immunopathology. Our study reveals that sustained type I interferon (IFN) signaling is a driver of pathogenic neutrophil responses and identifies IL-18 as a novel component of disease during genital HSV-2 infection.

## Introduction

Neutrophils are a critical component of the innate immune system. In humans, they are the most abundant leukocytes in circulation and are often among the first wave of immune cells responding to pathogen invasion. In the context of bacterial or fungal infections, including those that are sexually transmitted, neutrophils are largely protective and can help eliminate pathogens through a variety of effector functions, including phagocytosis, production of reactive oxygen species (ROS), neutrophil extracellular traps (NET) and protease release, and cytokine and chemokine secretion (*Mayadas et al., 2014*; *Pham, 2006*; *Tecchio et al., 2014*). In contrast, the role of neutrophils during viral infection is less clear (*Galani and Andreakos, 2015*). While neutrophils have been reported to neutralize several viruses and display protective qualities in vivo (*Akk et al., 2016*; *Jenne et al., 2013*; *Saitoh et al., 2012*; *Tate et al., 2009*; *Tate et al., 2011*), they have also been associated with

**eLife digest** Herpes simplex virus (HSV) is a human pathogen that causes genital herpes, an incurable disease that results in recurrent sores and inflammation. Infection with HSV induces a strong antiviral immune response, which results in large numbers of immune cells arriving at these lesions. But while some of these cells help to control viral replication, others might contribute to the inflammation that drives the disease.

One of the first immune cells to respond to infection are neutrophils. Although neutrophils are generally protective, especially against bacteria and fungi, they have also been implicated in tissue damage and severe inflammation during viral infections. But what determines whether a neutrophil will help to fight off an infection or increase disease severity is still an open question.

To investigate this, Lebratti, Lim et al. studied mice that had been infected with the genital herpes virus HSV-2, which is known to cause significant amounts of inflammation in mice. The experiments revealed that a signaling molecule called type I interferon, which is thought to be antiviral, causes neutrophils at the site of the infection to produce proteins, such as IL-18, which trigger an inflammatory reaction. Lebratti, Lim et al. found that type I interferon and IL-18 had shifting roles during the course of infection. In the early stages, both molecules had a protective effect, confirming results from previous studies. However, as the infection progressed, sustained levels of type I interferon signaling in neutrophils led to excess amounts of IL-18.

Lebratti, Lim et al. discovered that blocking interferon signaling or decreasing the levels of IL-18 later during infection unexpectedly reduced the severity of the disease and resulted in less genital tissue damage. Further experiments also showed that mice infected with another genital herpes virus called HSV-1 did not experience sustained levels of type I interferon. This may explain why this virus causes less severe disease in mice.

Understanding how the immune system reacts to viruses could reveal new targets for treatments of genital herpes. At the moment, there is little information about IL-18 production during genital herpes in humans. So, the next step is to see whether neutrophils behave in the same way and whether IL-18 can be detected during human disease. It is possible that the same immune components could promote disease in other infections too. If so, this work may help uncover new drug targets for other viral diseases.

---

tissue damage, loss of viral control, and increased mortality (*Bai et al., 2010*; *Brandes et al., 2013*; *Kulkarni et al., 2019*; *Narasaraju et al., 2011*; *Vidy et al., 2016*).

Type I interferons (IFNs) are potent regulators of neutrophil activity in a multitude of contexts. Type I IFNs can enhance recruitment of neutrophils to sites of infection, regulate neutrophil function, and drive immunopathology after infection by different classes of pathogens, including *Plasmodium spp.*, *Candida spp.*, and *Pseudomonas spp.* (*Majer et al., 2012*; *Pylaeva et al., 2019*; *Rocha et al., 2015*). However, type I IFNs can also inhibit neutrophil recruitment to the ganglia by suppressing chemokine expression after herpes simplex virus (HSV) infection (*Stock et al., 2014*), suggesting that the interplay of IFNs and neutrophil activity may be dependent on tissue type and the pathogen. The relationship between neutrophil-intrinsic type I IFN signaling and infection outcomes is less clear. Type I IFNs can promote expression of interferon-stimulated genes (ISGs) and pro-inflammatory cytokines in neutrophils, suggesting a potential role for them in driving immunopathology (*Galani et al., 2017*). During Leishmania infection, however, IFNAR signaling appears to suppress neutrophil-dependent killing of parasites (*Xin et al., 2010*), which emphasizes the complexity of IFN-mediated neutrophil responses.

Genital herpes is a chronic, sexually transmitted infection that affects over 400 million people worldwide (*World Health Organization, 2007*) and can be caused by two members of the alphaherpesvirus family, HSV-2 or the related HSV-1. Genital herpes is characterized by recurrent episodes of inflammation and ulceration, and the factors that drive disease are unclear. In humans, ulcer formation is associated with suboptimal viral control and spread during episodes of reactivation (*Roychoudhury et al., 2020*; *Schiffer and Corey, 2013*; *Schiffer et al., 2013*), while in mouse models, severity of disease often correlates with susceptibility to infection and the level of viral replication in the genital mucosa (*Gopinath et al., 2018*). Neutrophil infiltration into sites of active HSV-2

ulcers has also been reported in humans (*Boddingius et al., 1987*), but whether these cells are helpful or harmful during HSV infection is unknown. While neutrophils have been associated with tissue damage after multiple routes of HSV-1 infection (*Divito and Hendricks, 2008*; *Khoury-Hanold et al., 2016*; *Rao and Suvas, 2019*; *Thomas et al., 1997*), a protective role for neutrophils after genital HSV-2 infection has also been reported (*Milligan, 1999*; *Milligan et al., 2001*), although the use of non-specific depletion antibodies has muddled the respective contribution of neutrophils and other innate immune cells such as monocytes, which are known to be antiviral (*Iijima et al., 2011*). Furthermore, increased neutrophil recruitment to the HSV-2-infected vaginal epithelial barrier resulted in greater epithelial cell death, suggesting that neutrophil responses may indeed be pathogenic (*Krzyzowska et al., 2014*). However, the factors that distinguish pathogenic vs. non-pathogenic neutrophil responses during viral infection, including HSV-2 infection, remain ill-defined.

To address this, we evaluated the impact of neutrophils on genital disease severity using two models of HSV infection that result in low levels (HSV-1) or high levels of inflammation (HSV-2) (*Lee et al., 2020*). Between these two states, heightened expression of type I IFN during the resolution phase of acute infection and sustained expression of ISGs in neutrophils were detected after HSV-2 infection but not HSV-1. Therapeutic antibody-mediated blockade of IFNα/β receptor 1 (IFNAR1) as well as neutrophil-specific deletion of IFNAR1 reduced both genital inflammation as well as vaginal IL-18 levels during the resolution phase of acute HSV-2 infection. Accordingly, therapeutic neutralization of IL-18 also ameliorated genital disease after HSV-2 infection. Together, our data demonstrates that sustained type I IFN signaling is a key determinant of pathogenic neutrophil responses during viral infection, and identifies neutrophil- and type I IFN-dependent IL-18 production as a novel driver of inflammation during genital HSV-2 infection.

## Results

### Neutrophils are a component of severe genital inflammation after vaginal HSV-2 infection

To determine the role of neutrophils in our model of vaginal HSV-2 infection, wild-type (WT) female C57BL/6 mice were treated with Depo-Provera (depot medroxyprogestrone, DMPA) to hold mice at the diestrus phase of the estrus cycle and normalize susceptibility to infection (*Kaushic et al., 2003*). Neutrophils were depleted in DMPA-treated mice by intraperitoneal (i.p.) injection of an antibody against Ly6G, a neutrophil-specific marker, or an isotype control. One day later, mice were inoculated intravaginally with 5000 plaque forming units (PFU) of WT HSV-2 strain 186 syn+ (WT HSV-2). Neutrophils were effectively reduced up to 6 days post-infection (d.p.i.) in the vagina (*Figure 1A*) and the blood (*Figure 1—figure supplement 1*). In order to focus on genital inflammation, mice were monitored for 1 week after infection, as progression of disease within the second week of our infection model is largely indicative of viral dissemination into the central nervous system. In both cohorts, mild genital inflammation was apparent starting at 4 d.p.i. in a small fraction of mice (*Figure 1B*). Over time, progression of disease in the neutrophil-depleted mice was significantly slower compared to the controls. Remarkably, as late as 7 d.p.i., a proportion of the neutrophil-depleted group remained uninflamed, in contrast to the isotype control group in which all mice displayed signs of inflammation (*Figure 1B*). To confirm the disparity in disease severity, we examined the vagina and genital skin by histology. At 6 d.p.i., epithelial denuding and damage was apparent in the isotype control-treated mice (*Figure 1C*). In contrast, only a limited amount of epithelial destruction was observed in neutrophil-depleted mice, with less cellular infiltrates at sites of damage and in the lumen (*Figure 1C*). Furthermore, the epithelial layer proximal to areas of damage was morphologically distinct in isotype control-treated animals compared to neutrophil-depleted animals, suggesting diverse epithelial responses after infection in the presence or absence of neutrophils (*Figure 1C*). Similarly, destruction of the epidermis and separation of the epidermis from the dermis were widespread in the genital skin of isotype control-treated mice but not in neutrophil-depleted mice (*Figure 1C*). Unexpectedly, differences in genital inflammation and mucosal damage were largely independent of changes in viral control in the absence of neutrophils, as viral shedding into the vaginal lumen (*Figure 1D*) and viral control in the tissue parenchyma (*Figure 1E*) were similar between the two groups. Indeed, disease severity was decreased in neutrophil-depleted mice despite a slight delay in the resolution of viral replication at 5 d.p.i. (*Figure 1D*).

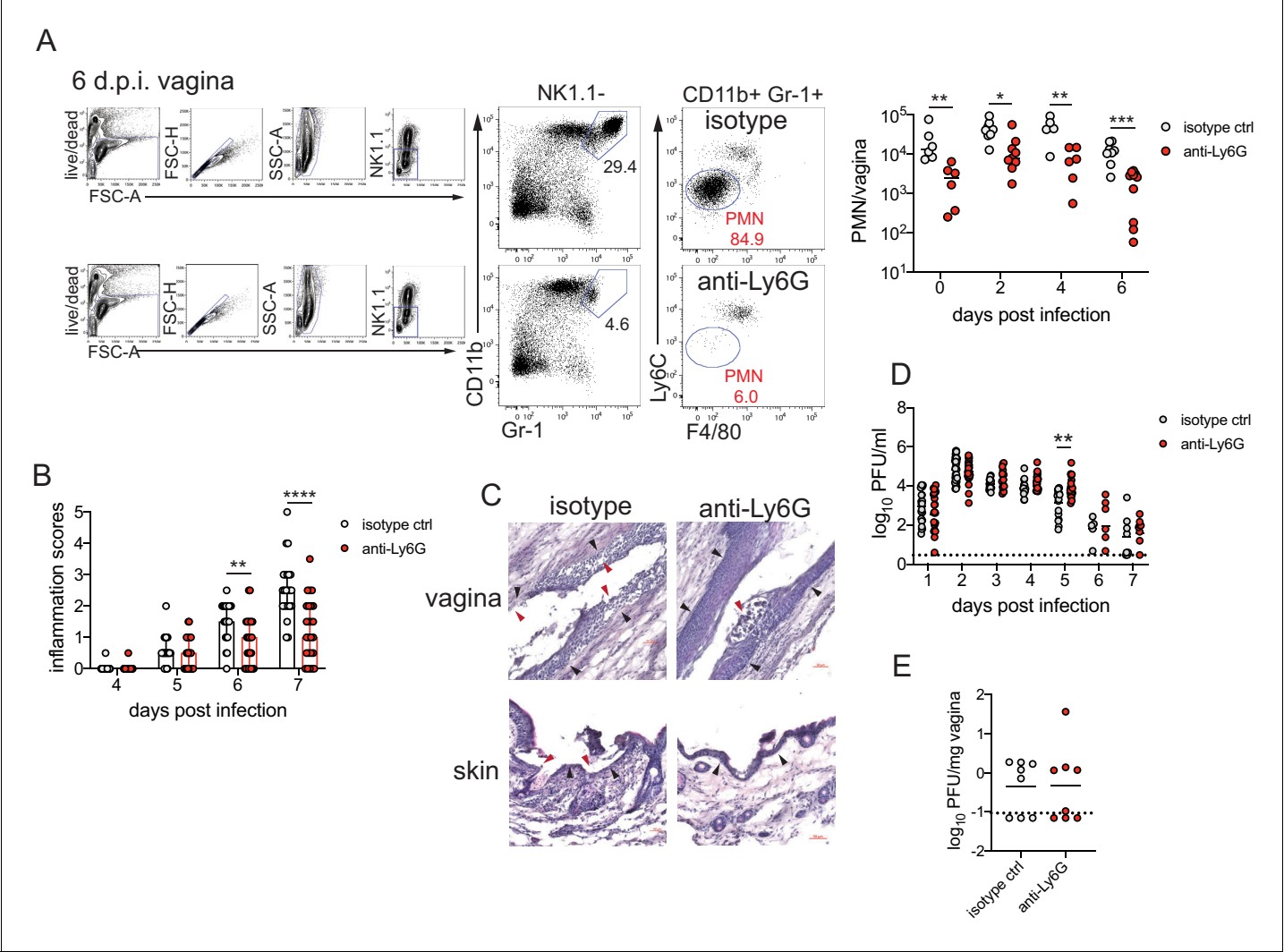

**Figure 1.** Neutrophil depletion reduces disease severity after HSV-2 vaginal infection. Female C57BL/6J mice were treated with depot medroxyprogestrone (DMPA) and inoculated intravaginally (ivag) with 5000 plaque forming units (PFU) of herpes simplex virus-2 (HSV-2). One day prior to HSV-2 inoculation, mice were injected intraperitoneally (i.p.) with 500 μg of rat IgG2a isotype control or anti-Ly6G monoclonal antibody (mAb). (**A**) Plots show gating strategy to identify neutrophils in the vagina. Numbers in plots refer to percent of parent population for gated cells (d0, d2: n = 6, d2: n = 9, d6 isotype: n = 8, d6 anti-Ly6G: n = 10). Depletion was confirmed by flow cytometry in the vagina on the indicated days. (**B**) Inflammation scores over the first 7 d.p.i. of mice treated with anti-Ly6G antibody (n = 25) or isotype control (n = 23). Mice showed no signs of disease prior to 4 d.p.i. (**C**) Histology of the vagina (top) or genital skin (bottom) at 6 d.p.i. from isotype control (left) or anti-Ly6G antibody-treated mice (right). Red arrows point to areas of epithelial denuding or damage, black arrows denote the basement membrane. (**D**) Infectious virus as measured by plaque assay in vaginal washes collected daily (both groups d1: n = 22, d2: n = 28, d3: n = 15, d4: n = 16, d5: n = 19, d6: n = 6, d7: n = 8). (**E**) Viral load was measured in homogenized vaginal tissue collected at 7 d.p.i. from the indicated groups (n = 8). Data in A and E are pooled from two independent experiments, and data in B and D are pooled from two to four independent experiments. Data in C is representative of two independent experiments. Bars in B show median with interquartile range. Horizontal bars in A, D, and E show mean. Scale bars show 50 mm. Statistical analysis was performed by two-way ANOVA on log-transformed data with Bonferroni's multiple comparisons test (**A**), repeated measures two-way ANOVA with Geisser-Greenhouse correction and Bonferroni's multiple comparisons test (**B**) repeated measures two-way ANOVA with Bonferroni's multiple comparison's test (**D**) or Mann-Whitney test (**E**). *p<0.05, **p<0.01, ***p<0.005, ****p<0.001. Raw values for each biological replicate, epsilon values, and specific p values are provided in *Figure 1—source data 1*.

The online version of this article includes the following source data and figure supplement(s) for figure 1:

**Source data 1.** Excel file with individual inflammation scores, viral titers, tissue weights, description of statistical tests, epsilon values and actual p values for *Figure 1*.

**Source data 2.** Excel file with individual cell numbers and frequencies, inflammation scores, viral titers, description of statistical tests, epsilon values and actual p values for *Figure 1—figure supplements 1–4*.

**Figure supplement 1.** Depletion of neutrophils in the blood after anti-Ly6G mAb treatment.

*Figure 1 continued on next page*

*Figure 1 continued*

**Figure supplement 2.** Neutrophil depletion does not affect magnitude of the immune cell response after HSV-2 infection.
**Figure supplement 3.** PAD4 is not required for development of genital inflammation during HSV-2 infection.
**Figure supplement 4.** ROS production and STIM1/STIM2 expression in neutrophils are not required for genital inflammation after HSV-2 infection.

We next evaluated whether the decreased inflammation after neutrophil depletion was due to changes in the cellular response against HSV-2 infection. We examined the recruitment of Ly6C + monocytes, NK cells, and CD4 and CD8 T cells (*Figure 1—figure supplement 2A*), all of which have been implicated in either the control of HSV or modulation of disease severity (*Lee and Ashkar, 2012*; *Shin and Iwasaki, 2013*; *Truong et al., 2019*). To remove intravascular cells and to limit our analysis to cells within the vagina, tissues were thoroughly perfused prior to collection (*Scott et al., 2018*). Unexpectedly, there was no significant difference in the number of Ly6C + CD11b + cells (*Figure 1—figure supplement 2B*), NK cells (*Figure 1—figure supplement 2C*), total CD4 (*Figure 1—figure supplement 2D*), or CD8 T cells (*Figure 1—figure supplement 2E*) that were recruited to the vagina over the first 6 days after infection regardless of whether neutrophils were present or not. Thus, our data demonstrate that neutrophils do not play a significant antiviral role in our model of vaginal HSV-2 infection, and rather promote genital inflammation with minimal impact viral burden and recruitment of other immune cells to the vagina.

## Neutrophil extracellular trap formation and oxidative burst are not major drivers of genital inflammation after HSV-2 infection

We next wanted to determine whether neutrophil-specific effector functions were promoting disease after HSV-2 infection. NETs have been associated with tissue damage in the context of both infectious (*Jenne and Kubes, 2015*) and non-infectious disease (*Granger et al., 2019*). To test whether NETs play a role in genital disease after HSV-2 infection, we first examined the ability of neutrophils to form NETs when exposed to HSV-2. In vitro stimulation of neutrophils with HSV-2 resulted in the enlargement of cell nuclei and the characteristic expulsion of DNA coated in citrullinated histones, which is a key characteristic of NETs (*Figure 1—figure supplement 3A*). The formation of NETs requires input from multiple pathways, including histone citrullination by enzymes such as PAD4, which leads to chromatin de-condensation and the eventual release of DNA (*Li et al., 2010*). To generate animals that were specifically lacking PAD4 in neutrophils, we bred *Padi4*$^{fl/fl}$ x *S100a*8-Cre mice (PAD4 CKO). HSV-2 infection of these mice and their littermate controls demonstrated minimal impact on genital inflammation (*Figure 1—figure supplement 3B*) or viral replication (*Figure 1—figure supplement 3C*) in the genital mucosa. Thus, our data show that PAD4 expression in neutrophils and likely NET formation are not the mechanisms by which these cells mediate disease after HSV-2 infection.

We next tested whether ROS production by neutrophils mediated inflammation after HSV-2 infection. While production of ROS in neutrophils supports antimicrobial activity against a variety of pathogens (*Dinauer, 2019*), excessive oxidative stress can be associated with tissue injury (*Mittal et al., 2014*). We found that in vitro stimulation of neutrophils with HSV-2 led to an increase in ROS production compared to unstimulated cells (*Figure 1—figure supplement 4A*). To determine whether respiratory burst in neutrophils promoted genital inflammation after HSV-2 infection in vivo, we infected mice with germline deficiency in *Ncf* (Ncf2 KO), which encodes p67$^{phox}$, a key component of the NADPH oxidase complex (*Jacob et al., 2017*). HSV-2 infection of Ncf2 KO and heterozygous controls resulted in similar progression of disease (*Figure 1—figure supplement 4B*) and did not alter viral titer (*Figure 1—figure supplement 4C*). To confirm that tested neutrophil effector functions, including ROS production, had little impact on genital inflammation, we infected mice in which the calcium-sensing molecules STIM1 and STIM2 were deleted from neutrophils, as these calcium-sensing molecules cooperatively regulate neutrophil activation and select effector functions (*Clemens et al., 2017*). *Stim1*$^{fl/fl}$ x S*tim2*$^{fl/fl}$ x *S100a8*-Cre (STIM1/2 DKO) mice were infected with HSV-2 and monitored for disease. As expected, there was little difference in genital inflammation severity between the STIM1/2 DKO and Cre- controls (*Figure 1—figure supplement 4D*) or viral titers (*Figure 1—figure supplement 4E*). Together, our data show that ROS production

from neutrophils and other cell types play little role in driving genital inflammation after HSV-2 infection.

## A type I IFN signature distinguishes neutrophil responses after genital HSV-1 and HSV-2 infection

To identify the factors that drove pathogenic neutrophil responses after HSV-2 infection, we turned to a complementary model of HSV-1 genital infection that we had previously described (*Lee et al., 2020*). Inoculation with the same dose of HSV-1 and HSV-2 led to profound differences in genital inflammation (*Figure 2A*) despite comparable levels of mucosal viral shedding throughout most days after infection (*Figure 2—figure supplement 1A*; *Lee et al., 2020*), although resolution of

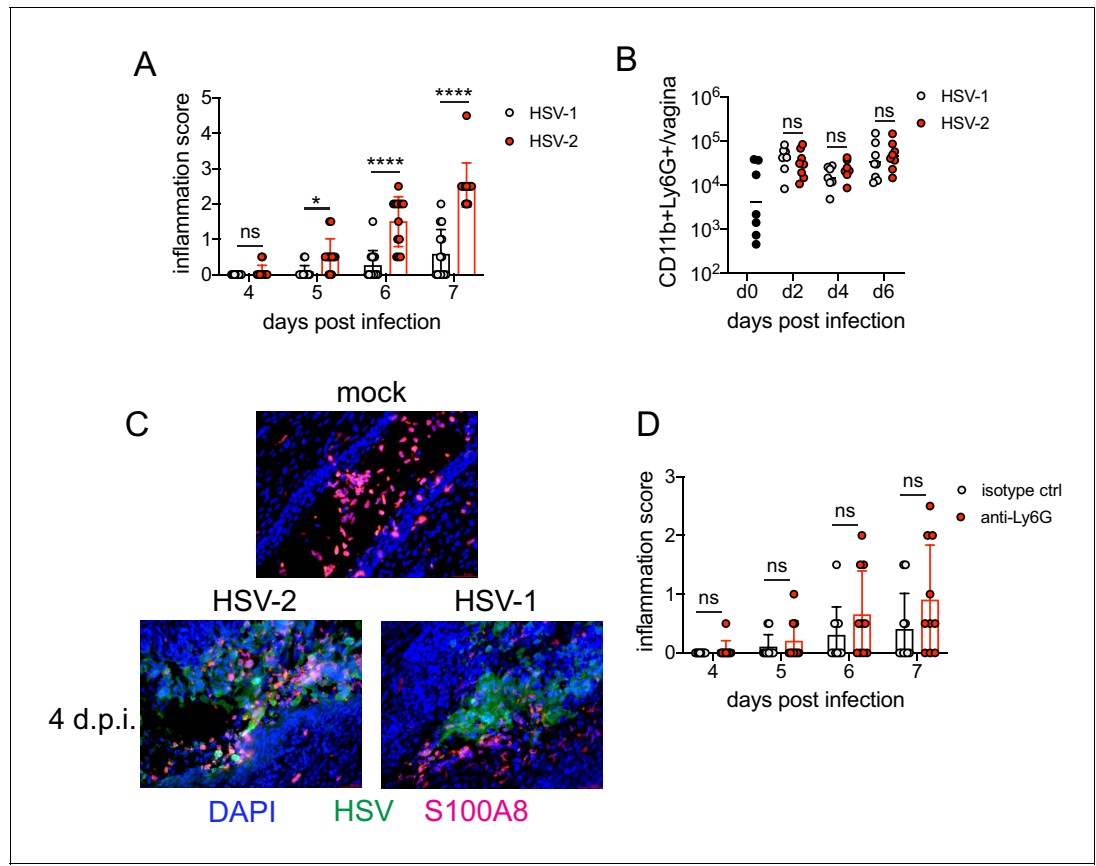

**Figure 2.** Neutrophils are non-pathogenic in a less inflammatory model of vaginal HSV-1 infection. Female C57BL/6J mice were treated with DMPA and inoculated ivag with $10^4$ PFU HSV-1 McKrae or HSV-2. (**A**) Inflammation scores were monitored for 7 d.p.i. (HSV-1: n = 14, HSV-2: n = 13). (**B**) Neutrophils were counted by flow cytometry in vaginal tissues at the indicated days after HSV-1 or HSV-2 infection (d0: n = 7, day 2: n = 8, day 4: n = 7, day 6: n = 8). (**C**) Vaginas were harvested from phosphate buffered saline (PBS)-inoculated (mock), HSV-1- or HSV-2-inoculated mice at 4 d.p.i., and tissue sections were probed with antibodies against HSV proteins (green) or S100A8 (red). 4',6-diamidino-2-phenylindole (DAPI) (blue) was used to detect cell nuclei. Images are representative of six mice per group. (**D**) Mice were treated with isotype control or anti-Ly6G mAb as described in *Figure 1* and then inoculated ivag with $10^4$ PFU HSV-1 McKrae. Inflammation scores were monitored for 7 d.p.i. (isotype, anti-Ly6G mAb: n = 10). Data are pooled from three (**A**) or two (**B–D**) independent experiments. Data in C is representative of two independent experiments. Bars show median with interquartile range (**A, D**) or mean (**B**). Scale bars show 50 mm. Statistical significance was measured by repeated measures two-way ANOVA with Geisser-Greenhouse correction and Bonferroni's multiple comparisons test (**A, D**) or two-way ANOVA with Bonferroni's multiple comparisons test (**B**). *p<0.05, ****p<0.001, ns = not significant. Raw values for each biological replicate, epsilon values, and specific p values are provided in *Figure 2— source data 1*.

The online version of this article includes the following source data and figure supplement(s) for figure 2:

**Source data 1.** Excel file with individual inflammation scores, cell numbers, description of statistical tests, epsilon values and actual p values for *Figure 2*.
**Figure supplement 1.** Neutrophil depletion prior to HSV-1 genital infection has minimal impact on viral control.
**Figure supplement 1—source data 1.** Excel file with individual viral titers, tissue weights, description of statistical tests and actual p values for *Figure 2—figure supplement 1*.

HSV-2 infection at 6 and 7 d.p.i. was delayed (*Figure 2—figure supplement 1A,B*). Importantly, magnitude of the neutrophil response in the vagina was similar between HSV-1- and HSV-2-infected mice during the course of acute mucosal infection (*Figure 2B*), and neutrophils could be found infiltrating sites of both HSV-1- and HSV-2-infected epithelium (*Figure 2C*). In contrast to HSV-2

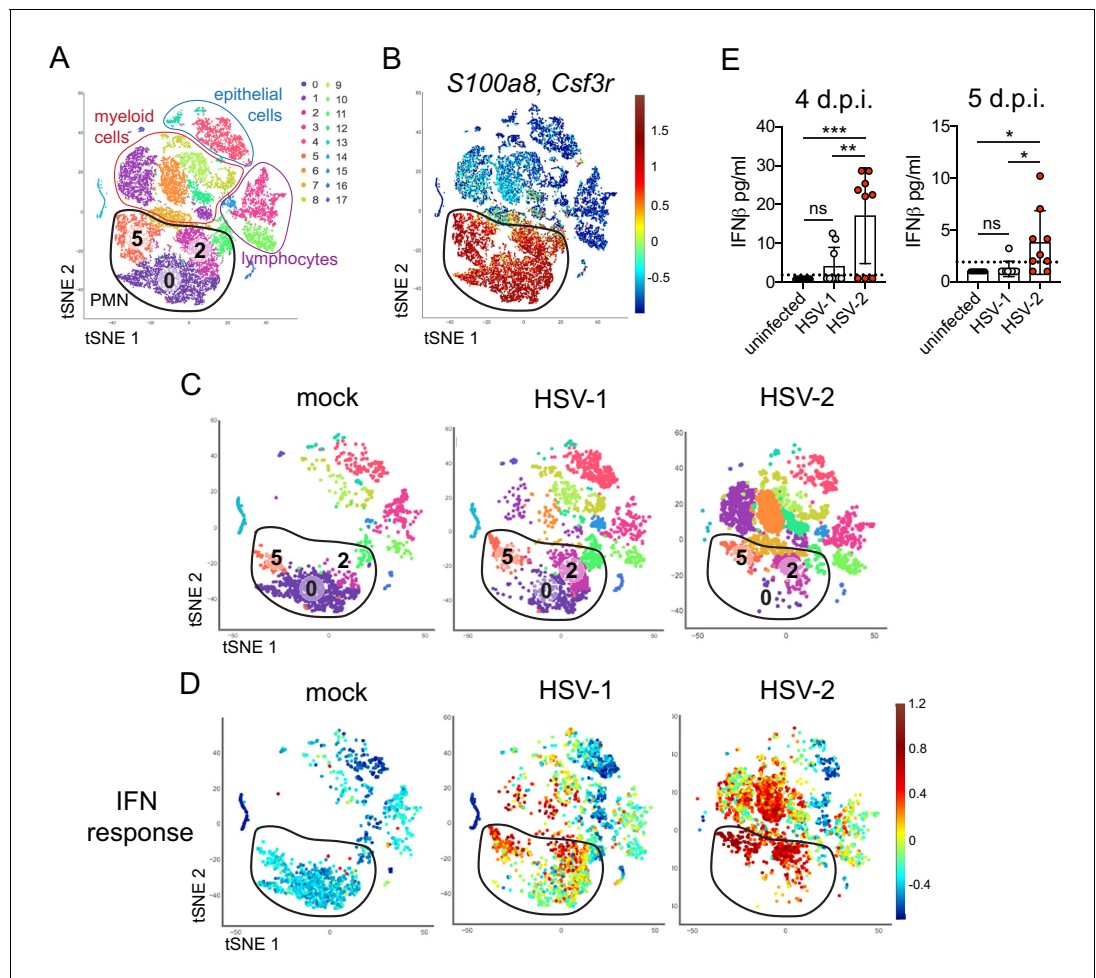

**Figure 3.** Single cell transcriptome analysis reveals a sustained IFN signature in the neutrophil response against HSV-2. Mice were infected as described in *Figure 2*. Vaginas were harvested at 5 d.p.i., and live cells were flow sorted and subjected to high-throughput single-cell RNA sequencing (scRNA-seq). (**A**) A t-Distributed Stochastic Neighbor Embedding (tSNE) visualization of 21,633 cells across all mice resolves 17 distinct clusters in the vaginal tissue. Clusters can be identified as myeloid cells (red border), epithelial cells (blue border), or lymphocytes (purple border). Neutrophils are encircled in black and contain three distinct clusters (0, 2, and 5). (**B**) Neutrophils are defined by high expression of *S100a8* and *Csf3r* (G-SCFR). (**C**) tSNE plots of vaginal cell clusters from mock-inoculated or HSV-infected mice. Neutrophil populations are circled in black. (**D**) Distribution of expression for genes within the Hallmark IFNa Response gene set. (**E**) Production of IFNβ in vaginal washes collected at 4 and 5 d.p.i. as measured by ELISA (uninfected: n = 8, HSV-1: n = 9, HSV-2: n = 9). scRNA-seq in A-D was performed once. Data in E are pooled from two independent experiments. Statistical significance was measured by one-way ANOVA with Tukey's multiple comparisons test. *p<0.05, **p<0.01, ***p<0.005. Raw values for each biological replicate, specific p values, and complete lists of differentially expressed genes between clusters 0, 2, and 5 are provided in *Figure 3—source data 1*.

The online version of this article includes the following source data and figure supplement(s) for figure 3:

**Source data 1.** Excel file with individual ELISA measurements, lists of differentially expressed genes between different neutrophil clusters, description of statistical tests and actual p values for *Figure 3*.

**Source data 2.** Excel file with lists of differentially expressed genes between neutrophils between HSV-1 and HSV-2 infected mice, individual delta Cq values, ELISA measurements, description of statistical tests and actual p values for *Figure 3—figure supplements 1–3*.

**Figure supplement 1.** ISGs are differentially expressed in neutrophils during HSV-1 and HSV-2 infection.

**Figure supplement 2.** Validation of ISG expression in the vagina.

**Figure supplement 3.** Type I IFN is produced early in the vagina and resolves by 1 week after acute HSV-1 or HSV-2 infection.

infection, antibody-mediated depletion of neutrophils with anti-Ly6G antibody prior to inoculation with HSV-1 did not reduce the development of genital inflammation during the first 7 days after infection (*Figure 2D*) and had minimal impact on acute viral control (*Figure 2D - Figure 2—figure supplement 1C*). Together, our data suggests that the regulation of the neutrophil response after HSV-1 or HSV-2 infection was distinct, which may contribute to the differences in disease outcomes between these infections.

To better understand the differences between pathogenic neutrophil responses after HSV-2 infection and the non-pathogenic neutrophil responses after HSV-1, we performed single-cell RNA sequencing (scRNA-seq) on sorted live vaginal cells from a mock-infected mouse or mice infected with HSV-1 or HSV-2 using the 10x Genomics platform (*Zheng et al., 2017*). Each sample was composed of cells from a single animal to better delineate potential subsets within cell populations, particularly neutrophils. Analysis across 21,633 cells in all samples revealed 17 unique clusters in the vagina during HSV infection after filtering, including myeloid cells, lymphocytes, and epithelial cells (*Figure 3A*). Neutrophils were identified by expression of known cell markers such as *S100a8* and *Csf3r* (*Figure 3B*). In mock-infected animals, the vaginal neutrophil population was dominated by cluster 0, and upon infection, at least two additional neutrophil subsets, cluster 2 and cluster 5, were clearly present (*Figure 3C*). While HSV-1-infected mice retained all three subpopulations of neutrophils in the vagina at 5 d.p.i., in HSV-2-infected mice, the presence of cluster 0 was greatly reduced, and the bulk of the neutrophils was composed of cluster 2 and 5 (*Figure 3C*). One major distinguishing characteristic between 'homeostatic' cluster 0 and 'infection' clusters 2 and 5 was the extent of ISG expression, in which cluster 0 expressed low levels of genes associated with a type I IFN response, even in infected animals, while clusters 2 and 5 expressed high levels of these genes (*Figure 3D*; *Liberzon et al., 2015*). Furthermore, the gene expression profile of clusters 2 and 5 was different between HSV-2 and HSV-1 infection at 5 d.p.i. (*Figure 3—figure supplement 1*), including the expression of ISGs (*Figure 3C*, *Figure 3—figure supplement 1*). Differential expression of select ISGs was confirmed by quantitative reverse transcription PCR (qRT-PCR) analysis in the vagina at 5 days after HSV-1 or HSV-2 infection (*Figure 3—figure supplement 2*). qRT-PCR shows that expression of CXCL10 (*Figure 3—figure supplement 2A,B*) and Gbp2 (*Figure 3—figure supplement 2C, D*; *Glennie et al., 2015*) is increased in HSV-2-infected vaginas compared to HSV-1, while IL-15 is not (*Figure 3—figure supplement 2E,F*), which supports the accompanying scRNA-seq analysis. While type I IFN was robustly produced early during acute infection after both HSV-1 and HSV-2 infection (*Figure 3—figure supplement 3*), IFNβ levels were higher in the vaginal lumen after HSV-2 infection compared to HSV-1 at time points corresponding to the onset of genital inflammation (*Figure 3E*) despite similar viral burden between the two infection models (*Figure 2—figure supplement 1A*). IFNβ was undetectable in both the vaginal lumen and the parenchyma by 7 days after both HSV-1 and HSV-2 infection (*Figure 3—figure supplement 3B,C*). Thus, during viral infection, distinct neutrophil subsets can be classified by transcriptional profiling, and expression of ISGs suggests that a key difference between a pathogenic and non-pathogenic neutrophil response during viral infection may be sustained IFN production and signaling.

## Sustained cell-intrinsic type I IFN signaling is required for pathogenic neutrophil responses during HSV-2 infection

We next wanted to test whether type I IFN signaling promoted immunopathology during genital HSV-2 infection. IFNAR1-deficient mice are highly susceptible to HSV, regardless of the route of inoculation (*Gill et al., 2006*; *Iversen et al., 2010*; *Iversen et al., 2016*; *Reinert et al., 2012*; *Royer et al., 2019*; *Svensson et al., 2007*; *Wilcox et al., 2016*), and rapidly succumb to infection, mainly due to a loss of viral control. To investigate the temporal effects of type I IFNs in HSV-2 genital disease, we used an antibody against IFNAR1 to block the receptor at different time points after infection (*Scott et al., 2018*). When mice were injected i.p. with anti-IFNAR1 antibody on the day of HSV-2 inoculation, disease progression was more rapid compared to isotype control-treated animals (*Figure 4—figure supplement 1A*), and the mice succumbed to infection at a faster rate (*Figure 4—figure supplement 1B*), in a manner similar to IFNAR1-deficient mice (*Iversen et al., 2010*; *Iversen et al., 2016*; *Lee et al., 2017*; *Reinert et al., 2012*; *Wang et al., 2012*). Inflammation and rapid disease progression were likely due to significantly elevated viral burden in the anti-IFNAR1 antibody-treated mice compared to isotype controls (*Figure 4—figure supplement 1C*), as HSV is a highly lytic virus that is capable of independently inducing epithelial tissue damage (*Horbul et al.,*

*2011*). To focus on the effects of persistent IFN signaling in the vagina after HSV-2 infection, we also treated mice with a single injection of anti-IFNAR1 antibody or an isotype control at 4 d.p.i. In stark contrast to early anti-IFNAR1 antibody treatment, one treatment with therapeutic IFNAR1 blockade led to a significant reduction in the severity of inflammation compared to isotype controls (*Figure 4A*). Histology of vaginal tissues from isotype-treated controls at 6 d.p.i. showed widespread epithelial denuding and immune cell infiltrates within the epithelial layer of the vagina (*Figure 4B*). In contrast, damage to the vaginal epithelium in anti-IFNAR1 antibody-treated mice appeared to be localized (*Figure 4B*), similar to neutrophil-depleted mice (*Figure 1C*). Similarly, the genital skin of isotype control-treated mice displayed signs of severe inflammation and destruction of the epidermis, while the skin structure of anti-IFNAR1 antibody-treated mice was largely intact (*Figure 4B*). Furthermore, IFNAR1 blockade at 4 d.p.i. had little impact on mucosal viral shedding (*Figure 4C*).

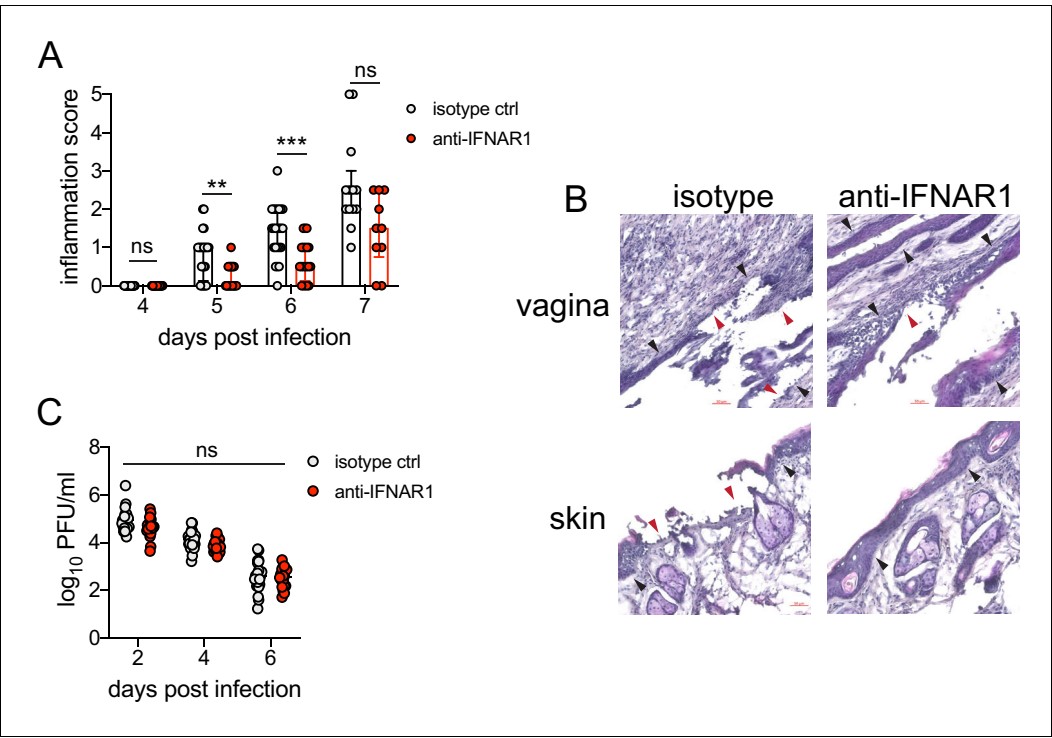

**Figure 4.** Inhibition of type I IFN signaling during the resolution phase of infection reduces inflammation after HSV-2 infection. Mice were infected as described in *Figure 2*. At 4 d.p.i., mice were injected i.p. with either 1 mg of anti-IFNAR1 antibody (n = 10–13) or isotype control (n = 7–9) and monitored for disease progression. Mice showing overt signs of genital inflammation at the time of antibody injection (4 d.p.i.) were excluded from the study. (A) Inflammation scores of antibody-treated mice over the first 7 d.p.i. (B) Histology of the vagina (top) or genital skin (bottom) at 6 d.p.i. Red arrows point to areas of epithelial denuding or damage. Black areas denote the basement membrane. (C) Infectious virus as measured by plaque assay in vaginal washes collected on the indicated days. Data are pooled from (A, C) or representative of three independent experiments. Bars in A show median with interquartile range; bars in C show mean. Scale bars show 50 mm. Statistical significance was measured by repeated measures two-way ANOVA with Geisser-Greenhouse correction and Bonferroni's multiple comparisons test (A) or two-way ANOVA with Bonferroni's multiple comparisons test (C). *p<0.05, **p<0.01, ***p<0.005, ns = not significant. Raw values for each biological replicate, epsilon values, and specific p values are provided in *Figure 4—source data 1*.

The online version of this article includes the following source data and figure supplement(s) for figure 4:

**Source data 1.** Excel file with indiviual inflammation scores, viral titers, description of statistical tests, epsilon values and actual p values for *Figure 4*.

**Figure supplement 1.** Early blockade of IFNAR1 leads to accelerated disease and loss of viral contol after HSV-2 infection.

**Figure supplement 1—source data 1.** Excel file with individual inflammation scores, raw survival data, viral titers, description of statistical tests, epsilon values and actual p values for *Figure 4—figure supplement 1*.

Collectively, these data show that the protective effect of type I IFN on control of genital HSV infection is limited to the early stages of acute infection, and that sustained IFN signaling in the later stages of acute HSV-2 genital infection drives inflammation with minimal effect on viral replication.

Single-cell transcriptional profiling data suggested that type I IFN signaling was robust in vaginal neutrophils after HSV-2 infection (*Figure 3D*). To determine whether intrinsic IFN signaling in neutrophils promoted immunopathology, we deleted IFNAR1 from granulocytes by breeding *Ifnar1*<sup>fl/fl</sup> x *S100a8*-Cre mice (IFNAR1 CKO). After confirming that IFNAR1 ablation was limited to the neutrophil population (*Figure 5A*), IFNAR1 CKO mice and littermate Cre- controls were vaginally infected with HSV-2. Despite differences in IFNAR1 expression, the number of neutrophils recovered from the vaginal lumen was similar between the IFNAR1 CKO mice and their Cre- control littermates

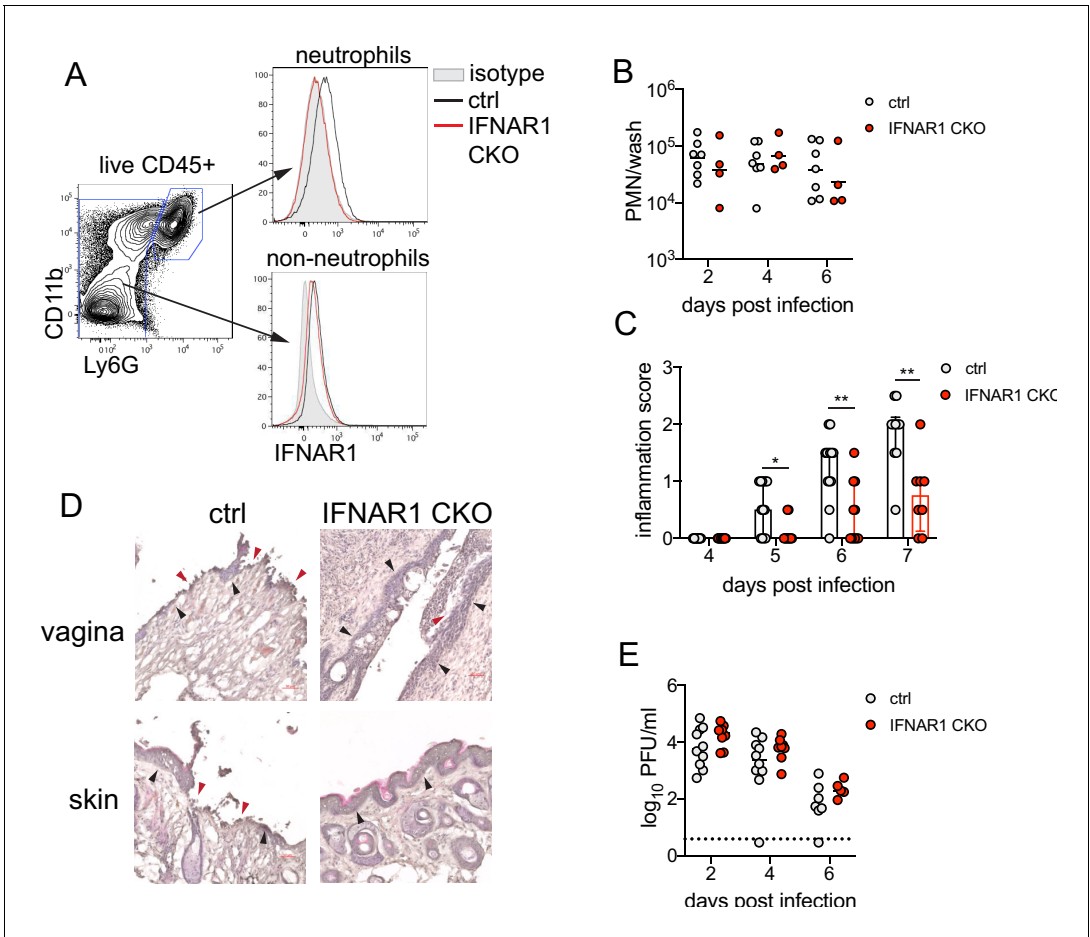

**Figure 5.** Type I IFN signaling in neutrophils promotes genital inflammation after HSV-2 infection. (A) IFNAR1 expression on neutrophils and non-neutrophil hematopoietic cells from the bone marrow of naive *Ifnar1*<sup>fl/fl</sup> x *S100a8-Cre* (IFNAR1 CKO) or Cre- littermate controls. Plot is gated on live CD45 + cells. CD11b + Ly6G + cells are neutrophils, Ly6G- cells are non-neutrophils. Gray histogram shows isotype staining, black open histogram is Cre- control, and red open histogram is IFNAR1 CKO. (B) Neutrophils were counted by flow cytometry in vaginal washes collected at the indicated days from IFNAR1 CKO (n = 4) or Cre- controls (n = 7) that were infected with HSV-2 as described in *Figure 1*. (C) Inflammation scores for the first 7 d.p.i. of IFNAR1 CKO (n = 10–13) or Cre- controls (n = 8–11). (D) Histology of the vagina and genital skin at 6 d.p.i. Red arrows point to areas of epithelial denuding or damage, black arrows denote basement membrane (E) Infectious virus as measured by plaque assay from vaginal washes collected on the indicated days from IFNAR1 CKO (n = 5–8) or Cre- controls (n = 7–10). Data in C and E are pooled from three independent experiments; data in B are pooled from two independent experiments; and data in D are representative of two independent experiments. Bars in C show median with interquartile range, bars in B and E show mean. Scale bars show 50 mm. Statistical significance was measured by mixed-effects analysis with (C) or without (B, E) Geisser-Greenhouse correction and Bonferroni's multiple comparisons test. *p<0.05, **p<0.01, ns = not significant. Raw values for each biological replicate, epsilon values and specific p values are provided in *Figure 5—source data 1*.

The online version of this article includes the following source data for figure 5:

**Source data 1.** Excel file with individual cell numbers, inflammation scores, viral titers, description of statistical tests, epsilon values and actual p values for *Figure 5*.

(*Figure 5B*). Strikingly, although the magnitude of the vaginal neutrophil response was similar, we found that the severity of genital inflammation presented by the IFNAR1 CKO mice was significantly reduced compared to the Cre- controls (*Figure 5C*). As observed after neutrophil depletion, a subset of the IFNAR1 CKO cohort did not develop any signs of inflammation as late as 7 d.p.i. (*Figure 5C*). Similar to our observations with therapeutic IFNAR1 blockade, IFNAR1 CKO mice exhibited less pathology in both the vagina and genital skin compared to Cre- controls (*Figure 5D*). Distinct disease outcomes between the Cre- controls and IFNAR1 CKO mice occurred independently of viral control, as viral loads in the mucosa were similar between the two groups (*Figure 5E*). Together, our data demonstrates that tissue inflammation during HSV-2 infection is largely driven by prolonged type I IFN production, which acts directly upon neutrophils to drive disease.

## Sustained type I IFN signaling and neutrophils regulate production of pathogenic IL-18 in the vagina during HSV-2 infection

Type I IFN stimulation of neutrophils can upregulate ISGs as well as several pro-inflammatory cytokines (*Galani et al., 2017*). To determine whether type I IFN was driving disease by shaping the cytokine milieu within the vagina, we first measured several pro-inflammatory cytokines in the vagina at 5 d.p.i., in the presence or absence of neutrophils. The production of inflammatory cytokines such as IL-6 (*Figure 6—figure supplement 1A*), IL-1β (*Figure 6—figure supplement 1B*), or TNF (*Figure 6—figure supplement 1C*), all of which have been associated with genital inflammation and HSV-2 infection in humans (*Gosmann et al., 2017*; *Masson et al., 2014*; *Murphy and Mitchell, 2016*), was similar between both neutrophil-depleted and control groups. Production of IFNγ (*Figure 6—figure supplement 1D*) as well as IL-12p70 (*Figure 6—figure supplement 1E*), both cytokines associated with a type I immune response and important for HSV control, was similar between the neutrophil-depleted and control groups. However, when we measured IL-18, an IL-1 family cytokine that is primarily known for mediating innate defense (*Harandi et al., 2001*) and for promoting IFNγ production from NK cells during genital HSV-2 infection (*Lee et al., 2017*), we detected a notable difference between neutrophil-depleted and control mice (*Figure 6A*), suggesting an unexpected role for this cytokine in driving disease during HSV-2 infection.

To determine whether type I IFN signaling regulated IL-18 production in the vagina, we assessed IL-18 levels in the vaginal lumen after therapeutic antibody-mediated IFNAR1 blockade. At 5 d.p.i., similarly to neutrophil-depleted mice, we found that IL-18 levels were markedly reduced (*Figure 6B*). Importantly, measurement of IL-18 in the vagina of IFNAR1 CKO at 5 d.p.i. also revealed a significant decrease in cytokine levels compared to littermate controls (*Figure 6C*).

To determine whether IL-18 was playing a key role in driving immunopathology during genital HSV-2 infection, we therapeutically administered an IL-18-neutralizing antibody directly at the site of infection to HSV-2-infected animals starting at 3 d.p.i. in order to promote sufficient antibody concentration and activity in the relevant tissue. Remarkably, neutralization of IL-18 led to a considerable reduction in disease severity (*Figure 6D*), without any impact on viral control (*Figure 6E*). To determine the source of pathogenic IL-18 in the vagina, we probed vaginal tissues for the neutrophil marker S100A8 and IL-18 at 6 d.p.i. (*Figure 6F*). Detection of IL-18 and S100A8 around a single nucleus demonstrated that neutrophils could be a source of IL-18 during vaginal HSV-2 infection (*Figure 6F*). However, we also identified IL-18-reactive cells that were negative for S100A8 but in close proximity to neutrophils (*Figure 6F*), suggesting the potential for multiple cellular sources of IL-18. Thus, our data demonstrate that sustained type I IFN signaling in neutrophils leads to the production of vaginal IL-18 and reveal IL-18 to be a novel regulator of disease after HSV-2 infection.

## Discussion

In this study, we evaluated drivers of a pathogenic neutrophil response using a mouse model for an important human infection. We found that neutrophils promote genital inflammation without affecting antiviral activity after genital HSV-2 infection, suggesting that the neutrophil response is primarily immunopathogenic. Depletion of neutrophils led to a significant decrease in disease severity without altering recruitment of other immune cells or the production of common pro-inflammatory cytokines, and deficiency in genes controlling neutrophil effector functions such as ROS production and NET formation had little impact on progression of disease. Comparative analysis of single-cell transcriptional profiles revealed a strong type I IFN signature that was sustained in neutrophils responding to

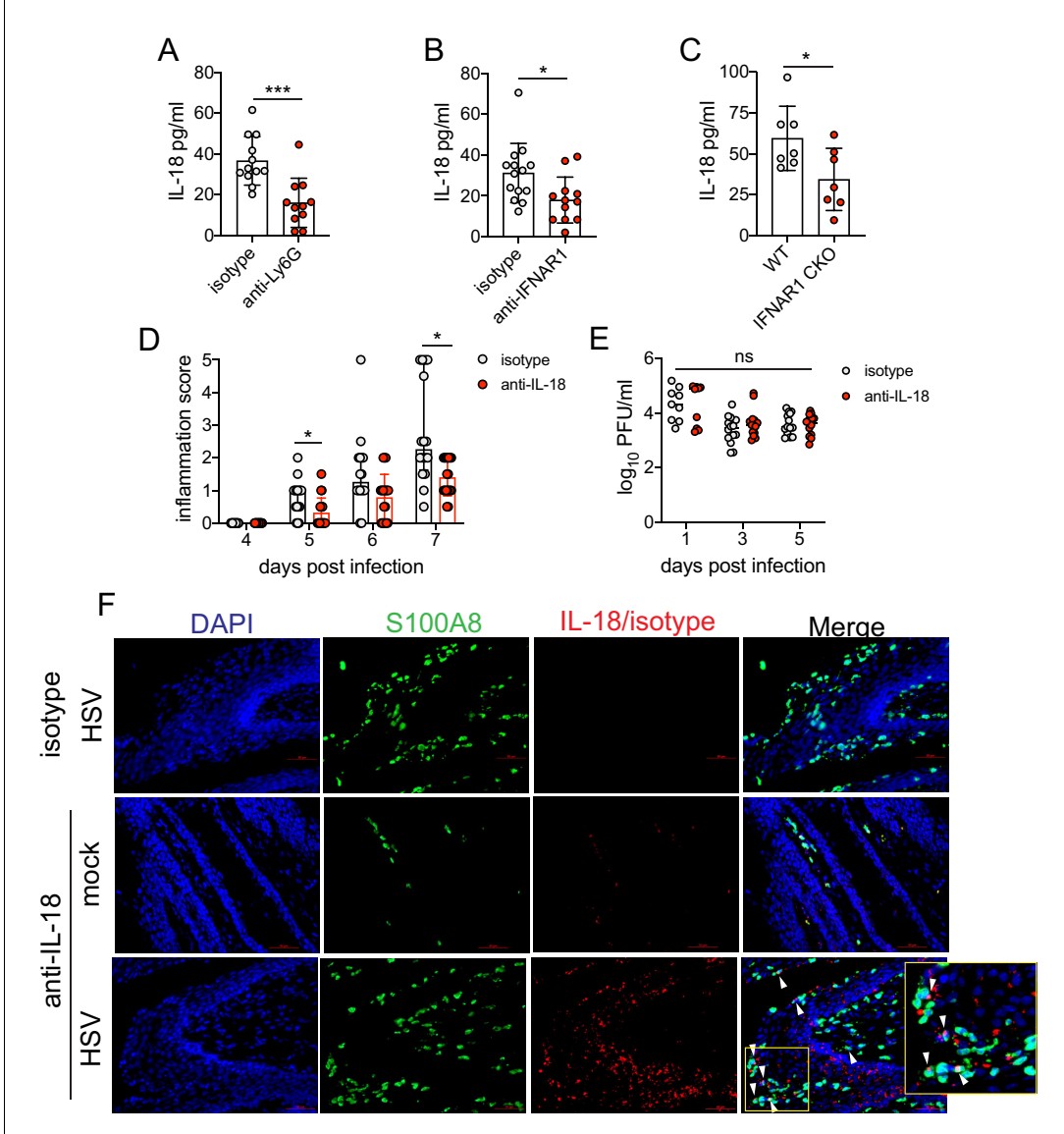

**Figure 6.** Sustained type I IFN signaling and neutrophils regulate pathogenic IL-18 levels in the vagina. C57BL/6 mice were treated with anti-Ly6G (n = 11) or isotype control (n = 12) as described in *Figure 1* (A), therapeutically treated with anti-IFNAR1 (n = 12) or isotype control (n = 14) as described in *Figure 5* (B) and infected with HSV-2, or IFNAR1 CKO (n = 7) and Cre- controls (n = 7) were infected with HSV-2 as described in *Figure 5* (C). (A-C) Vaginal IL-18 levels were measured by ELISA in washes collected at 5 d.p.i. 100 µg of anti-IL-18 neutralizing antibody (n = 18) or isotype control (n = 16) was administered ivag at 3, 4, and 5 d.p.i. (D) Inflammation scores of antibody-treated mice over the first 7 d.p.i. (E) Infectious virus as measured by plaque assay in vaginal washes collected on the indicated days (n = 9–14). (F) Immunofluorescent staining of vaginal tissues collected at 6 d.p.i. or from mock-inoculated mice. Green shows S100A8 (neutrophil), red shows IL-18 or isotype, blue is DAPI. Top row shows HSV-2-infected tissue probed with isotype control, and bottom two rows show representative images of tissues probed with anti-IL-18 antibody (middle = mock-infected, bottom = HSV-2-infected). White arrows point to IL-18 + neutrophils. Yellow square demarcates area shown in the inset. Data in A-C, D, and E are pooled from four independent experiments for each experimental setup. Data in F is representative of two independent experiments. Bars in A-C show mean and SD, bars in D show median with interquartile range, and bars in E show mean. Scale bars show 50 mm. Statistical significance was measured by unpaired t-test (A–C), repeated measures two-way ANOVA with (D) Geisser-Greenhouse correction and Bonferroni's multiple comparisons test, or mixed-effects analysis with Bonferroni's multiple comparisons test (D). *p<0.05, ***p<0.005, ns = not significant. Raw values for each biological replicate, epsilon values, and specific p values are provided in *Figure 6—source data 1*.

The online version of this article includes the following source data and figure supplement(s) for figure 6:

**Source data 1.** Excel file with individual ELISA measurements, inflammation scores, viral titers, description of statistical tests, epsilon values and actual p values for *Figure 6*.

**Figure supplement 1.** Neutrophils do not control production of common pro-inflammatory and antiviral cytokines during HSV-2 infection.

*Figure 6 continued on next page*

*Figure 6 continued*

**Figure supplement 1—source data 1.** Excel file with individual Bioplex assay values, description of statistical tests and actual p values for *Figure 6—figure supplement 1*.

a highly inflammatory genital HSV-2 infection but not a less inflammatory HSV-1 infection, suggesting that host responses to these two related viruses established distinct inflammatory milieus with divergent effects on responding neutrophils. In contrast to antibody-mediated blockade of IFNAR1 at the time of infection, which led to significantly worse disease outcomes, IFNAR1 blockade just prior to the resolution phase of acute mucosal infection significantly delayed the progression of genital inflammation. Importantly, neutrophil-specific deficiency of IFNAR1 markedly reduced the severity of genital disease after HSV-2 infection, suggesting that persistent IFN signaling drove disease primarily by acting on neutrophils. Ultimately, this sustained type I IFN signaling in neutrophils promoted the production of pro-inflammatory IL-18, and therapeutic neutralization of this cytokine also ameliorated disease. Together, our results suggest an axis of type I IFNs, neutrophils, and IL-18 as the key driver of genital disease in a mouse model of HSV-2 infection, and that sustained type I IFN signaling is a key factor in distinguishing between pathogenic and non-pathogenic neutrophil responses during mucosal viral infection.

Type I IFNs are a frontline of defense against viral infection, but models of chronic viral infection, including lymphocytic choriomeningitis virus (LCMV) (*Teijaro et al., 2013*; *Wilson et al., 2013*), human immunodeficiency virus (HIV) (*Meier et al., 2009*; *Rotger et al., 2010*; *Sedaghat et al., 2008*; *Taleb et al., 2017*), and simian immunodeficiency virus (SIV) (*Harris et al., 2010*; *Jacquelin et al., 2009*), reveal the detrimental effect of overexuberant or sustained type I IFN signaling. Notably, prolonged IFN signaling during chronic viral infection can promote immunosuppression through multiple cellular and molecular mechanisms, and deletion or blockade of IFNAR1 during chronic LCMV infection can alleviate immunosuppression and enhance long-term viral control (*Cheng et al., 2017*; *Taleb et al., 2017*; *Teijaro et al., 2013*; *Wilson et al., 2013*). However, unlike the LCMV model, early blockade of type I IFN signaling led to more severe disease and a complete loss of viral control after HSV-2 infection, similar to infections performed on an IFNAR1-deficient genetic background (*Iversen et al., 2010*; *Iversen et al., 2016*; *Lee et al., 2017*; *Leib et al., 1999*; *Reinert et al., 2012*; *Wang et al., 2012*), indicating an early antiviral role (*Lee et al., 2017*; *Luker et al., 2003*). Rather, only therapeutic inhibition of sustained IFN signaling diminished disease without disrupting viral control, thus revealing a heretofore unappreciated, temporal division of the antiviral and immunopathological effects of type I IFN signaling during HSV-2 infection. The source of sustained type I IFN production that promotes immunopathology after genital HSV-2 infection is currently unknown. It is also unclear as to why type I IFN production is sustained during HSV-2, but not after genital HSV-1 infection. Our data indicates a slight delay in the control of HSV-2 infection compared to HSV-1 at 6 and 7 d.p.i., but this did not correlate with differences in IFNβ levels between HSV-1 and HSV-2 infection. Differences in viral dissemination to the nervous system (*Lee et al., 2020*) or function of viral proteins may account for disparities in type I IFN production and ultimately, disease severity. HSV encodes numerous proteins that can suppress type I IFN production and regulate the signaling pathways (*Christensen et al., 2016*; *Lin and Zheng, 2019*; *Melroe et al., 2004*), suggesting that production of type I IFN likely occurs from a cell type that is not directly infected. While plasmacytoid dendritic cells (pDC) are known as robust producers of type I IFN, they appear to have a limited role during genital HSV-2 infection (*Swiecki et al., 2013*), indicating an alternative source of type I IFN, such as conventional DCs (*Wilson et al., 2013*). In humans, type I IFN can be detected at active lesion sites during recurrent episodes (*Peng et al., 2009*; *Roychoudhury et al., 2020*), although levels do not correlate with restriction of viral replication (*Roychoudhury et al., 2020*). This raises the possibility that type I IFN induction may not be antiviral and could contribute to ulcer formation, although this hypothesis is yet to be tested. Human neutrophils from females are also reported to be hyper-responsive to type I IFNs (*Gupta et al., 2020*). Although clinical disease recurrence rates between men and women with genital herpes are similar (*Wald et al., 2002*), differences in neutrophil sensitivity to type I IFN may have implications for sex-dependent mechanisms of ulcer development.

Synergistic effects of cytokine signaling have been reported to be important for maximizing cellular responses to infection through the upregulation of cooperative or independent molecular programs (*Bartee and McFadden, 2013*) or through the cross-regulation of receptor signaling pathways (*Ivashkiv and Donlin, 2014*). Upon infection, the activity of neutrophils can be modulated strongly by multiple IFNs, in a variety of tissues. Type I (*Ank et al., 2008*), type II (*Iijima et al., 2008*; *Lee et al., 2020*), and type III IFNs (*Ank et al., 2008*; *Lee et al., 2020*) are all robustly produced during HSV-2 infection. While type I and type II IFNs are crucial for the control of HSV-2 replication, endogenous type III IFNs do not appear to affect either disease severity or viral control, although exogenous application of type III IFNs can reduce viral burden (*Ank et al., 2008*; *Ank et al., 2006*). Expression of the type III IFN receptor, IFNLR, is limited to very few cell types, including neutrophils and epithelial cells (*Blazek et al., 2015*; *Mahlakõiv et al., 2015*; *Sommereyns et al., 2008*). As epithelial cells are a major target for HSV-2 replication, dissecting the action of type III IFNs within the neutrophil and epithelial cell compartments may reveal a more detailed picture of the role type III IFNs play. The impact of simultaneous type I, II, and III IFN signaling on neutrophil function is currently unclear, and due to the importance of these molecules in controlling infection, cell-specific modifications of receptor expression will be required to better understand their impact on neutrophil function.

In vitro stimulation of neutrophils with type I IFN leads to the upregulation of many common ISGs as well as inflammatory genes, including IL-18 (*Galani et al., 2017*). Importantly, type I IFN may differentially regulate expression of IL-18 and IL-1β, another IL-1 family cytokine that depends on caspase-mediated cleavage for activation (*Zhu and Kanneganti, 2017*). It is unclear whether neutrophils are directly producing this cytokine in our model of infection and whether IL-18 production is dependent on inflammasome activation. As HSV also encodes proteins that can inhibit inflammasome activity (*Maruzuru et al., 2018*), one possibility is that IL-18 is produced by a cell type that is not productively infected with HSV, such as neutrophils. Alternatively, neutrophil proteases released in the extracellular space have been reported to cleave and activate proIL-1 cytokines that are secreted by other cells in a caspase-1-independent manner (*Clancy et al., 2018*; *Robertson et al., 2006*; *Sugawara et al., 2001*), suggesting a mechanism by which neutrophils may modulate IL-18 levels without directly secreting the cytokine themselves. Our data show that along with neutrophils, IL-18 was present in the epithelium in cells that are in close proximity to infiltrating neutrophils. Although we have not yet confirmed whether this detected IL-18 is bioactive, our data allude to multiple sources and mechanisms by which pathogenic IL-18 is produced during HSV-2 infection.

During HSV-2 vaginal infection, IL-18 stimulates NK cells to rapidly produce antiviral IFNγ (*Lee et al., 2017*), and is thought to be important for orchestrating a protective innate immune response. Accordingly, IL-18-deficient mice are more susceptible to HSV-2 infection (*Harandi et al., 2001*) and HSV-1 infection (*Fujioka et al., 1999*; *Reading et al., 2007*), presumably due to dysregulation of innate IFNγ production and loss of viral control. Our study reveals a novel aspect of IL-18 biology during HSV-2 infection, and that like type I IFN signaling, there may be a temporal component to the effects of IL-18 during HSV-2 infection. Currently, the mechanism by which IL-18 promotes disease during genital HSV-2 infection is unknown. In the gut, the role of IL-18 is balanced between protection and pathology (*Jarret et al., 2020*; *Nowarski et al., 2015*). The role of IL-18 during HSV-2 infection appears to be similarly complex, and further study will be required to identify the compartment on which IL-18 acts and the downstream effects of IL-18 signaling. Additionally, while our results demonstrate an important role for IL-18, the reduction in disease severity was not as profound as therapeutic IFNAR blockade in our HSV-2 model of infection. Considering the complex response elicited by type I IFN, our data suggest that other IL-18-independent, IFN-dependent mechanisms that promote genital inflammation are yet to be elucidated. Nevertheless, therapeutic neutralization of IL-18 reduced disease without altering viral titers in our model, suggesting that IL-18 does not have an impact on T-cell-dependent IFNγ production (*Milligan and Bernstein, 1997*; *Nakanishi et al., 2009*) or direct antiviral activity. As previous studies have shown that IL-18 is also dispensable for stimulating IFNγ from adaptive memory immune responses (*Harandi et al., 2001*), IL-18, along with type I IFN, may present attractive targets for therapeutics aiming to reduce inflammation during genital herpes.

## Materials and methods

### Mice

Six-week-old female C57BL/6J mice were purchased from Jackson Laboratories and rested for at least 1 week and infected at a minimum of 7 weeks of age. *Ncf2* KO mice and controls were provided by M.C. Dinauer (Washington University, St Louis) and generated as previously described (*Jacob et al., 2017*). *Stim1*fl/fl x *Stim2*fl/fl x *S100a8*-Cre mice were provided by G.A. Clemens (Washington University, St Louis) and were generated as previously described (*Clemens et al., 2017*). *Ifnar1*fl/fl mice (*Ifnar1*tm1Uka) were a gift from H.W. Virgin (*Kamphuis et al., 2006*; *Nice et al., 2016*). *Padi4*fl/fl mice (B6(Cg)-*Padi4*tm1.2Kmow/J) and *S100a8*-Cre (B6.Cg-Tg(*S100a8*-cre,-EGFP)1Ilw/J) were obtained from Jackson Laboratories and bred at Washington University School of Medicine. Cre- littermates generated from breeding pairs were used as controls. All mice were maintained on a 12 hr light/dark cycle with unlimited access to food and water. This study was carried out in accordance with the recommendations in the Guide for the Car and Use of Laboratory Animals of the National Institutes of Health.

### Cell lines and primary cells

Vero Cells (African green monkey kidney epithelial cells, ATCC) were cultured in Dulbeco's Modified Eagle Medium (Gibco) containing 1% fetal bovine serum (FBS, Corning) and maintained at 37°C with 5% $CO_2$. Cells were regularly tested for mycoplasma contamination, and all cells used for this study were mycoplasma-free. Primary neutrophils were isolated from the bone marrow (BM) of naive female C57BL/6J mice. A Histopaque gradient was used to isolate primary neutrophils for ROS assays, while a Percoll gradient was used for NET assays. For Histopaque isolation: 3 ml of Histopaque 1119 (Sigma-Aldrich) was overlaid with 3 ml of Histopaque 1077 (Sigma-Aldrich). A single-cell suspension of isolated BM cells in 1 ml of PBS was layered over the Histopaque gradient. Cells were centrifuged for 30 min at room temperature (RT), and neutrophils were collected from the bottom interface. For Percoll isolation: BM cells were resuspended in HBSS (Gibco) with 20 mM HEPES (Gibco) and layered over 6 ml of 62% Percoll solution (GE Healthcare). Cells were centrifuged for 30 min at RT, and neutrophils were collected from the bottom of the tube. All tissue culture experiments were performed under BSL2 containment.

### Viruses and virus quantification

WT HSV-2 186 syn+ (*Spang et al., 1983*) and HSV-1 McKrae (*Williams et al., 1965*) were propagated and titered on Vero cells as previously described (*Lee et al., 2020*). Briefly, for propagation of virus stocks, Vero cells were plated in T150 tissue culture flasks, inoculated at 0.01 MOI at 80% confluence, and incubated at 37°C. Infected cells were harvested 2–3 days after infection, resuspended in equal volumes of virus supernatant and twice-autoclaved milk, and sonicated. Lysed cells were aliquoted and used as viral stock. To titer, Vero cells were plated in six-well plates and inoculated with 10-fold serial dilutions of stock virus. After inoculation, overlay media with 20 µg/ml human IgG was added to each well and plates were incubated at 37°C for 2–3 days. To count, Vero cells were stained with 0.1% crystal violet. All tissue culture experiments were performed under BSL2 containment. For titration of virus in the vaginal lumen, 50 ul washes with sterile PBS were collected using a pipette and a sterile calginate swab, and diluted in 950 ul of ABC buffer (0.5 mM $CaCl_2$, 0.5 mM $MgCl_2$, 1% glucose, and 1% FBS in sterile PBS). For titration of virus from tissue, vaginas were harvested into pre-weighed tubes and flash frozen on dry ice. ABC buffer was added to weighed tissues, which were bead-homogenized and clarified by centrifugation. 10-fold serial dilutions of vaginal washes or tissue homgenate were titered by plaque assays on Vero cells (*Lee et al., 2020*).

### Mouse infection studies

All mice were injected subcutaneously in the neck ruff once with 2 mg of DMPA (Depo-Provera, Pfizer) 5–7 days prior to virus inoculation. For experiments in which neutrophils were depleted, mice were i.p. injected once with 500 µg of anti-Ly6G (clone 1A8) or rat IgG2a isotype control (anti-trinitrophenol +KLH) (Leinco Technologies) diluted in sterile PBS (Sigma-Aldrich) 1 day prior to inoculation. For experiments in which IFNAR blockade was conducted, mice were i.p. injected once with

1 mg of anti-IFNAR1 (clone MAR1-5A3) or mouse IgG1 isotype control (clone HKSP) (Leinco Technologies) on either the day of inoculation ('early') or at 4 d.p.i. ('late'). For 'late' treatments, only mice without overt signs of genital inflammation were chosen for antibody injection in both anti-IFNAR and isotype control groups to avoid biasing of results. For experiments in which IL-18 was neutralized, mice were treated intravaginally with 100 µg of anti-IL-18 antibody (clone YIGIF74-1G7) or rat IgG2a isotype control (clone 2A3) (BioXCell) on days 3–5 after infection. Selection of mice for isotype control or experimental antibody treatment was random. For intravaginal inoculation, a sterile calginate swab (McKesson) moistened with sterile PBS was used to gently disrupt mucous from the vaginal cavity. Stock virus was diluted in sterile PBS and either 5000 PFU or $10^4$ PFU virus was delivered into the vaginal cavity via pipette tip in a 10 µl volume. Mice were weighed and monitored for signs of disease for 1 week following infection in an unblinded manner and monitored for survival for 2 weeks. Genital inflammation was scored as follows: 0 – no inflammation, 1 – mild redness and swelling around the vaginal opening, 2 – fur loss and visible ulceration, 3 – severe ulceration and mild signs of sickness behavior (lack of grooming), 4 –hindlimb paralysis, and 5 – moribund.

## Vaginal tissue processing

All tissues were harvested from animals sedated with ketamine and xylazine and thoroughly perfused with a minimum of 15 ml of PBS. Vaginas were processed as follows: tissue was cut into pieces and digested for 15 min in a shaking water bath held at 37°C in a 0.5 mg/ml solution of Dispase II (Roche) in PBS. Tissues were then transferred to a solution of 0.5 mg/ml Collagenase D (Roche) and 15 µg/ml DNase I (Roche) in RPMI media (Gibco) supplemented with 10% FBS (Corning) and 1% pen/strep (Gibco) and digested for 25 min in a shaking water bath held at 37°C. 50 µl of sterile EDTA was added to each sample and incubated at 37°C for another 5 min. Tissues were then mechanically disrupted through a 70 um cell strainer into a single-cell suspension using a 3 ml syringe plunger. Tissues were washed with RPMI media with 1% FBS, centrifuged, and resuspended in 200 µl RPRM with 1% FBS and 1% pen/strep.

## Flow cytometry

Single-cell suspensions from vaginal tissues, or luminal cells collected in vaginal washes were plated in 96-well plates and incubated with Live/Dead Fixable Aqua Dead Cell Stain kit (Molecular Probes) for 15 min at room temperature (RT) in the dark. Cells were then incubated with Fc block (anti-CD16/32, Biolegend) for 15 min at RT in the dark. Surface staining was performed in FACS buffer (1% FBS and 0.02% sodium azide in PBS) on ice and in the dark using the following antibodies: CD3 (clone 145–2 C11), CD4 (clone GK1.5), CD8a (clone 53–6.7), CD11b (clone M1/70), CD45 (clone 30-F11), Gr-1 (clone RB6-8C5), Ly6C (clone HK1.4), Ly6G (clone 1A8), and NK1.1 (clone PK136). All antibodies were purchased from Biolegend. For surface staining of IFNAR1, cells were incubated with an anti-IFNAR1 antibody or a mouse IgG1 isotype control (Leinco Technologies) for 20 min at 37°C. Cells were washed and then surface staining of other markers proceeded as described above. Cell counts were performed by adding Precision Count Beads (Biolegend) to samples prior to flow cytometric acquisition. Samples were acquired on an LSR Fortessa (BD Biosciences) and analyzed by FlowJo (Treestar).

## Tissue immunofluorescent staining and immunohistochemistry

All tissues were harvested from animals sedated with ketamine and xylazine and thoroughly perfused with a minimum of 15 ml of PBS, followed by 15 ml of PLP fixative (0.01 M NaIO$_4$, 0.075 M lysine, 0.0375 M sodium phosphate, and 2% paraformaldehyde [PFA]) for immunofluorescent (IF) staining or 4% PFA for immunohistochemistry (IHC). Tissues were cryoprotected in 30% sucrose, frozen in OCT medium (Sakura), and cut into 7 um sections. Cryosections were blocked 5% bovine serum albumin (BSA), 5% goat serum (Jackson Immunoresearch), and 0.1% Triton-X in PBS for 1 hr at RT. HSV antigens were detected with a rabbit anti-HSV primary antibody (Dako), incubated overnight at 4°C, washed in PBS, and incubated for 1 hr at RT with a goat anti-rabbit IgG conjugated to AlexaFluor 488 (Life Technologies). S100A8 was detected with a rat anti-mouse S100A8 primary antibody (clone 63N13G5, Novus Biologicals) and a goat anti-rat IgG conjugated to AlexaFluor 568 (Life Technologies) in a similar manner. IL-18 was detected using a biotinylated rat anti-mouse IL-18 primary antibody (clone 93–10C, MBL International). Cryosections were blocked as described above

and then treated with the Avidin/Biotin Blocking Kit (Vector Laboratories) according to manufacturer's protocol. Endogenous peroxidases were quenched with a 2% hydrogen peroxide solution. Anti-mouse IL-18 or a rat IgG1 isotype control was incubated overnight at 4°C. The AlexaFluor 647 Tyramide Signal Amplification kit (Invitrogen) was used to visualize IL-18 and used according to manufacturer's protocol. DNA was visualized with 4',6-diamidino-2-phenylindole (DAPI) (Life Technologies). Sections were imaged with a Zeiss Cell Observer inverted microscope using a 40x objective, acquired with Zen software, and image brightness was adjusted using Photoshop (Adobe). For IHC, sections were probed with an anti-HSV antibody incubated overnight at 4°C (Dako), a donkey anti-rabbit IgG-HRP antibody (Jackson Immunoresearch) for 1 hr at RT and then enzymatically visualized by 3,3'-diaminobenzidine (DAB) enzyme reaction (Sigma-Aldrich). Sections were counterstained with hematoxylin and eosin, and images were captured using Zeiss ZEN software on a Zeiss Cell Observer inverted microscope witβh an Axiocam dual B/W and color camera with a 20x objective. Image brightness was adjusted using Photoshop (Adobe) and merged with Image J64 (NIH).

## RNA extraction and quantitative reverse transcription PCR

Harvested tissues were homogenized in RLT buffer (RNeasy Kit, Qiagen) with approximately 100 µl of sterile 1.0 mm zirconia/silica beads (Biospec Products) in a bead beater. Homogenized tissue samples were processed according to manufacturer's protocol using the RNeasy Mini Kit (Qiagen) and RNA quality and quantity was assessed on a Nanodrop (ThermoFisher). qRT-PCR was performed in 10 µl reactions using the iTaq Universal SYBR Green One-Step kit (Biorad) according to manufacturer's protocol.

## Single-cell RNA-sequencing preparation

Single-cell suspensions from digested vaginas were stained with Live/Dead Fixable Aqua Dead Cell Stain kit (Molecular Probes) for 15 min at RT in the dark. Live cells were sorted on BD FacsAria II housed in a BSL2 biosafety cabinet. A minimum of 16,000 cells were resuspended in PBS with 2% FBS and 0.2 U/µl RNase inhibitor at a concentration of 800–1400 cells/µl, submitted to McDonnell Genome Institute, and prepared for droplet-based 3' end scRNA-seq using the Chromium 3' v3 single-cell reagent kit as per manufacturer's protocol (10x Genomics). Library sequencing was performed on a NovaSeq S4 (Illumina).

## Cytokine measurement

For cytokine analysis by Bio-Plex Pro Mouse Cytokine 23-Plex Immunoassay (Bio-rad): 2 × 50 µl washes with sterile PBS were collected from the vaginal lumen using a pipette. Samples were centrifuged for 3 min at 13000*g to remove mucous and cells, and supernatants were added to 200 µl of ABC buffer. The assay was performed according to manufacturer instructions, and plates were read on a Luminex Bioplex 100 system (Biorad). For measurement of IL-18 or IFNβ, 2 × 50 µl washes with sterile PBS were collected from the vaginal lumen and centrifuged to remove mucous and cells. IL-18 was measured using the mouse IL-18 ELISA kit (MBL International) according to manufacturer's instructions at half-volumes, while IFNβ was measured using the LEGEND MAX Mouse IFNβ ELISA kit (Biolegend) according to manufacturer's instructions. For the measurement of IFNβ in tissue homogenates, vaginal tissues were collected in pre-weighed tubes and flash frozen on dry ice. NP40 lysis buffer (150 mM NaCl, 50 mM Tris, pH 8.0, 1% NP40 alternative) with protease inhibitor cocktail (Sigma Aldrich) was added to weighed tissue and bead-homogenized. Lysates were clarified by centrifugation and supernatants used for ELISA.

## In vitro neutrophil stimulation

To measure ROS production, isolated neutrophils were stimulated with heat-killed HSV-2 (56°C for 30 min) at an MOI of 5 for 16 hr at 37°C. ROS levels were quantified using DCFDA Cellular ROS Detection Assay kit (Abcam) according to manufacturer's protocol. Fluorescence levels were measured by flow cytometry. To induce NET formation, neutrophils were stimulated with heat-killed HSV-2 at an MOI of 1 for 4 hr at 37°C. Cells were fixed with 8% PFA overnight and probed with a polyclonal rabbit antibody against mouse citrullinated histone H3 (Abcam) for 1 hr at RT in 1% BSA and 0.1% Triton-X for 1 hr at RT, a goat anti-rabbit antibody conjugated to AlexaFluor 488 (Life Technologies) for 1 hr at RT and DAPI diluted in PBS for 6 min at RT. Cells were imaged with a Zeiss

Cell Observer inverted microscope using a 63x objective and image brightness was adjusted using Photoshop (Adobe).

## Single-cell RNA-sequencing analysis
### Processing data with Seurat package
The Seurat package in R was used for analysis (*Butler et al., 2018*). Cells with mitochondrial content greater than 5% were removed. The initial analysis of the data revealed three clusters of the cells that had extremely low levels of detected genes (i.e. <500), which were then filtered out as non-viable cells. Remaining cells were used for downstream analysis, resulting in the 6,507 cells per sample that passed quality control (QC) and filtering. Filtered data were normalized using a scaling factor of 10,000, and nUMI was regressed with a negative binomial model.

### Normalization and feature selection
After the data filtration, data were normalized using a scaling factor of 10,000 and log-transformed. The highly variable genes were selected using the FindVariableFeatures function with mean greater than 0.0125 or less than three and dispersion greater than 0.5. These genes were used in performing the linear dimensionality reduction.

### Clustering and finding markers
Principal component analysis was performed using the top 3000 most variable genes prior to clustering and number of the first principal components (PCs) was used based on the ElbowPlot as described below for different datasets. Clustering was performed using the FindClusters function, which works on K-nearest neighbor (KNN) graph model with the granularity (resolution) ranging from 0.1 to 1.5. The datasets were projected as t-SNE plots.

## Statistical analysis
All numerical data analyses except for scRNA-seq data analysis were performed on Graphpad Prism8 software. Values were log-transformed to normalize distribution and variances where necessary. Immune cell numbers and cytokine measurement were analyzed by two-way ANOVA with Bonferroni's multiple comparisons test. Log-transformed viral titers were analyzed by repeated measures two-way ANOVA with Bonferroni's multiple comparisons test. Inflammation scores were analyzed by repeated-measures two-way ANOVA or mixed-effects analysis with Geisser-Greenhouse correction and Bonferroni's multiple comparisons test. The Geisser-Greenhouse correction was used for inflammation scores to correct any violations of sphericity and to provide a more restrictive, stringent calculation of p values. ROS MFI was measured by unpaired two-tailed Student's t-test. qPCR results were analyzed by one-way ANOVA with Tukey's multiple comparisons test. A $p<0.05$ was considered statistically significant. No experimental data points were excluded from statistical analysis, including potential outliers. Mouse and sample numbers per group and experimental repeat information is provided in the figure legends. All data points represent individual biological replicates, and the 'n' for each group refers to biological replicates. No power calculations were performed to determine sample size; rather sample sizes were determined based on historical data.

## Acknowledgements
We thank Rachel Idol, Antonina Akk, and Celeste Cummings for technical assistance on assays used in this study. This work was supported by grants from the NIH (HS: R01 AI134962) and Children's Discovery Institute (RAC). TJL was supported by funding for the Training Program in Immunology from the NIH (T32 AI007163).

## Additional information

### Funding

| Funder | Grant reference number | Author |
|---|---|---|
| National Institutes of Health | R01 AI134962 | Haina Shin |

| National Institutes of Health | T32 AI007163 | Tania Lebratti |

The funders had no role in study design, data collection and interpretation, or the decision to submit the work for publication.

## Author contributions

Tania Lebratti, Maria Rita Fabbrizi, Formal analysis, Investigation; Ying Shiang Lim, Formal analysis, Investigation, Writing - original draft; Adjoa Cofie, Xiaoping Jiang, Jason Scott, Ayşe Naz Ozantürk, Investigation; Prabhakar Andhey, Formal analysis, Visualization; Christine Pham, Regina Clemens, Mary Dinauer, Resources, Writing - review and editing; Maxim Artyomov, Resources, Software, Supervision; Haina Shin, Conceptualization, Data curation, Supervision, Funding acquisition, Writing - original draft, Writing - review and editing

## Author ORCIDs

Maria Rita Fabbrizi (iD) https://orcid.org/0000-0002-5156-1575
Ayşe Naz Ozantürk (iD) http://orcid.org/0000-0002-0187-3642
Haina Shin (iD) https://orcid.org/0000-0002-1014-1451

## Ethics

Animal experimentation: This study was carried out in accordance with the recommendations in the Guide for the Car and Use of Laboratory Animals of the National Institutes of Health. The protocols were approved by the Institutional Animal Care and use Committee (IACUC) at the Washington University School of Medicine (Assurance number A3381-01). All experiments were performed under biosafety level 2 (A-BSL2) containment and all efforts were made to minimize animal suffering.

## Decision letter and Author response

Decision letter https://doi.org/10.7554/eLife.65762.sa1
Author response https://doi.org/10.7554/eLife.65762.sa2

# Additional files

## Supplementary files

• Transparent reporting form

## Data availability

Sequencing data have been deposited in GEO under accession code GSE161336.

The following dataset was generated:

| Author(s) | Year | Dataset title | Dataset URL | Database and Identifier |
|---|---|---|---|---|
| Fabbrizi MR, Lim YS, Andhey PS, Artyomov M, Shin H | 2020 | scRNAseq of mock, HSV-1 or HSV-2 infected C57BL/6 Vaginal tissue at 5 days post infection | https://www.ncbi.nlm.nih.gov/geo/query/acc.cgi?acc=GSE161336 | NCBI Gene Expression Omnibus, GSE161336 |

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

# Appendix 1

**Appendix 1—key resources table**

| Reagent type (species) or resource | Designation | Source or reference | Identifiers | Additional information |
|---|---|---|---|---|
| strain, strain background (Herpes Simplex Virus) | HSV-2 186 syn+ | PMID:6296440 | | |
| strain, strain background (Herpes Simplex Virus) | HSV-1 McKrae | PMID:14223669 | | |
| strain, strain background (*Mus musculus*) | C57BL/6J | Jackson Laboratory | Cat#000664; RRID:IMSR_JAX:000664 | |
| strain, strain background (*Mus musculus*) | *Ncf2* KO | PMID:28471497 (Provided by M. C. Dinauer) | N/A | |
| strain, strain background (*Mus musculus*) | *Stim1*$^{fl/fl}$ x *Stim2*$^{fl/fl}$ x *S100a8*-Cre | PMID:28724541 (Provided by G. A. Clemens) | N/A | |
| strain, strain background (*Mus musculus*) | *Padi4*$^{fl/fl}$ | Jackson Laboratory | B6(Cg)-Padi4tm1.2Kmow/ J, RRID:IMSR_JAX:026708 | |
| strain, strain background (*Mus musculus*) | *Ifnar1*$^{fl/fl}$ | PMID:27327515, PMID:16868248 | MGI:2655303 (Provided by H.W. Virgin) | |
| strain, strain background (*Mus musculus*) | *S100a8*-Cre | Jackson Laboratory | B6.Cg-Tg(S100A8-cre,-EGFP)1Ilw/J, RRID:IMSR_JAX:021614 | |
| strain, strain background (*Mus musculus*) | *Ifnar1*$^{fl/fl}$ x *S100a8*-Cre | This paper | MGI:2655303 (Provided by H.W. Virgin) and B6.Cg-Tg(S100A8-cre,-EGFP)1Ilw/J, RRID:IMSR_JAX:021614 | We crossed *Ifnar1*$^{fl/fl}$ mice obtained from H.W. Virgin with commercially available *S100a8*-Cre mice. |
| strain, strain background (*Mus musculus*) | *Padi4*$^{fl/fl}$ x *S100a8*-Cre | This paper | B6(Cg)-Padi4tm1.2Kmow/ J, RRID:IMSR_JAX:026708 and B6.Cg-Tg(S100A8-cre,-EGFP)1Ilw/J, RRID:IMSR_JAX:021614 | We crossed commercially available *Padi4*$^{fl/fl}$ mice with *S100a8-Cre* mice. |
| cell line (*Cercopithecus aethiops*) | Vero Cells (CCL81) | ATCC | (CCL-81; RRID:CVCL_0059) | |
| biological sample (*Mus musculus*) | Primary mouse bone marrow neutrophils | C57BL/6J (Jackson Laboratory) | N/A | Freshly isolated using a histopaque gradient. |
| antibody | anti-mouse Ly6G (1A8) (Rat monoclonal) | Leinco Technologies | Cat# L280, RRID:AB_2737551 | 500 μg/ mouse by intraperitoneal injection |
| antibody | anti-mouse IFNAR1 (MAR1-5A3) (Mouse monoclonal) | Leinco Technologies | Cat# I-401, RRID:AB_2491621 | 1mg/mouse by intraperitoneal injection |
| antibody | IgG2a (1-1) (Rat monoclonal) | Leinco Technologies | Cat# R1367, RRID:AB_2831721 | 500 μg/ mouse by intra peritoneal injection |

*Continued on next page*

*Appendix 1—key resources table continued*

| Reagent type (species) or resource | Designation | Source or reference | Identifiers | Additional information |
|---|---|---|---|---|
| antibody | IgG1 (HKSP84) (Mouse monoclonal) | Leinco Technologies | Cat# I-117, RRID:AB_2830510 | 1mg/mouse by intraperitoneal injection |
| antibody | BV605 anti-mouse CD3ε (145–2 C11) (Armenian Hamster monoclonal) | Biolegend | Cat# 100351, RRID:AB_2565842 | (1:200) |
| antibody | Pacific Blue anti-mouse CD4 (GK1.5) (Rat monoclonal) | Biolegend | Cat# 100427, RRID:AB_493646 | (1:200) |
| antibody | PE/Dazzle 594 anti-mouse CD4 (GK1.5) (Rat monoclonal) | Biolegend | Cat# 100455, RRID:AB_2565844 | (1:500) |
| antibody | APC anti-mouse CD8a (53–6.7) (Rat monoclonal) | Biolegend | Cat# 100712, RRID:AB_312751 | (1:800) |
| antibody | APC anti-mouse/human CD11b (M1/70) (Rat monoclonal) | Biolegend | Cat# 101211, RRID:AB_312794 | (1:2000) |
| antibody | PerCPCy5.5 anti-mouse Gr-1 (RB6-8C5) (Rat monoclonal) | Biolegend | Cat# 108427, RRID:AB_893561 | (1:400) |
| antibody | BV605 anti-mouse Gr-1 (RB6-8C5) (Rat monoclonal) | Biolegend | Cat# 108439, RRID:AB_2562333 | (1:100) |
| antibody | BV605 anti-mouse Ly-6C (HK1.4) (Rat monoclonal) | Biolegend | Cat# 128035, RRID:AB_2562352 | (1:600) |
| antibody | Pacific Blue anti-mouse Ly-6C (HK1.4) (Rat monoclonal) | Biolegend | Cat# 128013, RRID:AB_1732090 | (1:200) |
| antibody | PerCPCy5.5 anti-mouse Ly6G (1A8) (Rat monoclonal) | Biolegend | Cat# 127616, RRID:AB_1877271 | (1:400) |
| antibody | FITC anti-mouse NK1.1 (PK136) (Mouse monoclonal) | Biolegend | Cat# 108706, RRID:AB_313393 | (1:200) |
| antibody | AF700 anti-mouse CD45 (30-F11) (Rat monclonal) | Biolegend | Cat# 103127; RRID:AB_493714 | (1:1000) |
| antibody | PE anti-mouse IFNAR1 (MAR1-5A3) (Mouse monoclonal) | Leinco Technologies | Cat# I-1033-6717-1400, RRID:AB_2830368 | (1:20) |
| antibody | PE IgG1 (HKSP) (Mouse monoclonal) | Leinco Technologies | Cat# I-104-475-1800, RRID:AB_2830371 | (1:20) |
| antibody | anti-mouse CD16/32 (Fc block) (Rat monoclonal) | Biolegend | Cat# 101320, RRID:AB_1574975 | (1:100) |

*Continued on next page*

*Appendix 1—key resources table continued*

| Reagent type (species) or resource | Designation | Source or reference | Identifiers | Additional information |
|---|---|---|---|---|
| antibody | Live/Dead Fixable Aqua Dead Cell Stain kit | Molecular Probes | Cat# L34957 | (1:500) |
| antibody | anti-HSV primary antibody (Rabbit polyclonal) | Dako | Cat# B0116, RRID:AB_2335703 | (1:1500) |
| antibody | anti-mouse S100A8 (63N13G5) (Rat monoclonal) | Novus Biologicals | Cat# NBP2-27067 | (1:200) |
| antibody | anti-mouse biotinylated IL-18 (93–10C) (Rat monoclonal) | MBL International | Cat# D048-6, RRID:AB_592012 | (1:200) |
| antibody | biotinylated IgG1 Isotype Control (eBRG1) (Rat monoclonal) | Thermo Fisher Scientific | Cat# 13-4301-81, RRID:AB_470080 | (1:200) |
| antibody | anti-Histone H3 (citrulline R2) IgG (Rabbit polyclonal) | Abcam | Cat# ab80075, RRID:AB_1603562 | (1:200) |
| antibody | Anti-Rabbit IgG H and L (Alexa Fluor 488) (Goat Polyclonal) | Life Technologies | Cat# ab150077, RRID:AB_2630356 | (1:1000) |
| antibody | anti-Rat IgG H and L (Alexa Fluor 568) (Goat Polyclonal) | Life Technologies | Cat# A-11077, RRID:AB_2534121 | (1:1000) |
| antibody | anti-rat IgG-H and L (Alexa Fluor 488) (Donkey polyclonal) | Life Technologies | Cat# A-21208, RRID:AB_2535794 | (1:1000) |
| antibody | anti-rabbit IgG-HRP antibody (Donkey polyclonal) | Jackson Immunoresearch | Cat# 711-005-152, RRID:AB_2340585 | (1:1000) |
| antibody | Human Serum IgG-Fractionated Purified 1G | Innovative Research | Cat# 50643486 | 10 µg/mL |
| antibody | IgG2a Isotype control (2A3) (Rat monoclonal) | BioXCell | Cat# BE0089, RRID:AB_1107769 | 100 µg/ mouse by intravaginal administration |
| antibody | anti-mouse IL-18 (YIGIF74-1G7) (Rat monoclonal) | BioXCell | Cat# BE0237, RRID:AB_2687719 | 100 µg/ mouse by intravaginal administration |
| sequence-based reagent | Ifnb1_F | PMID:30559377, IDT | PCR Primers | 5'-AACCTCACC TACAGGGCGGAC TTCA-3' |
| sequence-based reagent | Ifnb1_R | PMID:30559377, IDT | PCR Primers | 5'-TCCCACGTCAA TCTTTCCTCTTGC TTT-3' |
| sequence-based reagent | Il15_F | IDT | PCR Primers | 5'-AGACTTGCAG TGCATCTCCTTA-3' |
| sequence-based reagent | Il15_R | IDT | PCR Primers | 5'-CTTTCCTGACC TCTCTGAGCTGTT-3' |

*Continued on next page*

*Appendix 1—key resources table continued*

| Reagent type (species) or resource | Designation | Source or reference | Identifiers | Additional information |
|---|---|---|---|---|
| sequence-based reagent | Cxcl10_F | IDT | PCR Primers | 5'-GATGACGGGCCAGTGAGAAT-3' |
| sequence-based reagent | Cxcl10_R | IDT | PCR Primers | 5'-TCGTGGCAATGATCTCAACA-3' |
| sequence-based reagent | Gbp2_F | PMID:26216123, IDT | PCR Primers | 5'-ACCTGGAACATTCCCTGACC-3' |
| sequence-based reagent | Gbp2_R | PMID:26216123, IDT | PCR Primers | 5'-ACAGCTCCTCCTCCCGCAGAG-3' |
| sequence-based reagent | Rpl13_F | IDT | PCR Primers | 5'-GCGGATGAATACCAACCCCT-3' |
| sequence-based reagent | Rpl13_R | IDT | PCR Primers | 5'-ACCACCATCCGCTTTTTCTTG-3' |
| commercial assay or kit | Bio-Plex Pro Mouse Cytokine 23-Plex Immunoassay | Bio-Rad | Cat# M60009RDPD, RRID:AB_2857368 | |
| commercial assay or kit | DCFDA/H2 DCFDA - Cellular ROS Assay Kit | Abcam | Cat# ab113851 | |
| commercial assay or kit | Fast DAB tablet set | Sigma-Aldrich | Cat# D4293-5SET | |
| commercial assay or kit | AlexaFluor 647 Tyramide Signal Amplification kit | Invitrogen | Cat# T20951 | |
| commercial assay or kit | LEGEND MAX Mouse IFNb ELISA kit | Biolegend | Cat# 439407 | |
| commercial assay or kit | Mouse IL-18 ELISA Kit | MBL International Corporation | Cat# 7625 | |
| commercial assay or kit | RNeasy Mini kit | Qiagen | Cat# 74106 | |
| commercial assay or kit | iTaq Universal SYBR one-step kit | Biorad | Cat# 172–5151 | |
| commercial assay or kit | 10 × 3' v3 Single Cell Library | 10x Genomics | | |
| chemical compound, drug | Paraformaldehyde | Fisher Scientific | Cat# 04042–500 | |
| chemical compound, drug | Sodium Azide | Sigma-Aldrich | Cat# S2002-25G | |
| chemical compound, drug | Calcium Chloride Dihydrate ($CaCl_2 \cdot 2H_2O$) | AMRESCO, Inc. | Cat# 0556–500G | |
| chemical compound, drug | Magnesium Chloride Hexahydrate ($MgCl_2 \cdot 6H_2O$) | Sigma-Aldrich | Cat# M9272-500G | |
| chemical compound, drug | Sodium (meta) periodate ($NaIO_4$) | Sigma-Aldrich | Cat# S1878-25G | |
| chemical compound, drug | Triton X-100 | Sigma-Aldrich | Cat# T8787-100ML | |
| chemical compound, drug | L-Lysine (Hydrochloride) | MP Biomedicals | Cat# 194697 | |

*Continued on next page*

*Appendix 1—key resources table continued*

| Reagent type (species) or resource | Designation | Source or reference | Identifiers | Additional information |
|---|---|---|---|---|
| chemical compound, drug | Sodium Phosphate monobasic monohydrate (NaH$_2$PO$_4$ · H$_2$O) | Sigma-Aldrich | Cat# S9638-500G | |
| chemical compound, drug | Sodium Phosphate Dibasic (Na$_2$HPO$_4$) | Sigma-Aldrich | Cat# 795410–500G | |
| chemical compound, drug | D-(+)-Glucose | Sigma-Aldrich | Cat# G5767-500G | |
| chemical compound, drug | Sucrose | Sigma-Aldrich | Cat# S5016-500G | |
| chemical compound, drug | Depot Medroxyprogestrone (DMPA, Depo-Provera) | Pfizer | Cat# 421035 | 2mg/mouse by subcutaneous injection. |
| chemical compound, drug | Dispase II | Roche | Cat# 4942078001 | |
| chemical compound, drug | Collagenase D | Roche | Cat# 11088866001 | |
| chemical compound, drug | DNase I | Roche | Cat# 10104159001 | |
| chemical compound, drug | Fetal Bovine Serum (FBS) | Corning | lot reserve# 35010125 | |
| chemical compound, drug | Dulbecco's Modified Eagle Medium (DMEM) | Gibco | Cat# 11965084 | |
| chemical compound, drug | RPMI-1640 Medium | Gibco | Cat# 11875093 | |
| chemical compound, drug | Hanks' Balanced Salt solution (HBSS) | Gibco | Cat# H6648-500ML | |
| chemical compound, drug | Dulbecco's Phosphate Buffered Saline (sterile use) | Sigma-Aldrich | Cat# D8537-500L | |
| chemical compound, drug | Dulbecco's Phosphate Buffered Saline (non-sterile use) | Sigma-Aldrich | Cat# D5652-10L | |
| chemical compound, drug | Penicillin-Streptomycin (10,000 U/mL) | Gibco | Cat# 15140122 | |
| chemical compound, drug | HEPES (1M) | Gibco | Cat# 15630080 | |
| chemical compound, drug | Percoll | GE Healthcare | Cat# GE17-0891-02 | |
| chemical compound, drug | Histopaque 1119 | Sigma-Aldrich | Cat# 11191–100 ML | |
| chemical compound, drug | Histopaque 1077 | Sigma-Aldrich | Cat# 10771–100 ML | |
| chemical compound, drug | bovine serum albumin | Sigma-Aldrich | Cat# A9418-10G | |
| chemical compound, drug | Normal goat serum | Jackson Immunoresearch | Cat# 005-000-121 | |
| chemical compound, drug | Normal donkey serum | Jackson Immunoresearch | Cat# 017-000-001 | |

*Continued on next page*

*Appendix 1—key resources table continued*

| Reagent type (species) or resource | Designation | Source or reference | Identifiers | Additional information |
|---|---|---|---|---|
| chemical compound, drug | NP40 substitute | Sigma-Aldrich | Cat# 492016 | |
| chemical compound, drug | Protease Inhibitor Cocktail | Sigma-Aldrich | Cat# P8340 | |
| software, algorithm | GraphPad Prism8 | GraphPad Software | RRID:SCR_002798 | http://www.graphpad.com/ |
| software, algorithm | FlowJo | FlowJo | RRID:SCR_008520 | https://www.flowjo.com/solutions/flowjo |
| software, algorithm | R | open source | RRID:SCR_001905 | https://www.r-project.org/ |
| software, algorithm | Cell Ranger | 10x Genomics | RRID:SCR_017344 | https://support.10xgenomics.com/single-cell-gene-expression/software/pipelines/latest/what-is-cell-ranger |
| software, algorithm | Seurat package | Satija lab | RRID:SCR_016341 | https://satijalab.org/seurat/get_started.html |
| software, algorithm | ZEN Digital Imaging for Light Microscopy Software | Zeiss | RRID:SCR_013672 | https://www.zeiss.com/microscopy/us/products/microscope-software/zen.html#introduction |
| software, algorithm | Photoshop | Adobe | RRID:SCR_014199 | https://www.adobe.com/products/photoshop.html |
| software, algorithm | Image J | National Institutes of Health | RRID:SCR_003070 | https://imagej.nih.gov/ |
| other | 4′,6-diamidino-2-phenylindole | Life Technologies | Cat# D1306 | |
| other | Hematoxylin | Thermo Scientific | Cat# 6765007 | |
| other | Eosin | Sigma-Aldrich | Cat# 102439 | |
| other | Dry Milk (Instant non fat) | Nestlé Carnation | N/A | |

