## [Decision Letter]

**Acceptance summary:**

This paper identifies three important novel points in regards to HSV pathogenesis in the vaginal mouse model. First, genital HSV-2 infection induces a greater degree of immunopathogenesis relative to genital HSV-1 despite no temporal differences in viral load or neutrophil quantity. Second, early disruption of the type 1 interferon inflammatory cascade has deleterious effects on genital HSV-2 infection while late disruption (as well as neutrophil depletion) spares mice from severe immunopathogenesis. Finally, therapeutic neutralization of IL-18 is also disease sparing in mice despite its ability to promote interferon γ production from NK cells. These findings may be of importance for human therapeutics but are also a reminder to a general immunology audience that helpful and harmful mucosal responses may differ for even slightly divergent viral pathogens and over time during an evolving infection. Congratulations on these important findings.

**Decision letter after peer review:**

Thank you for submitting your article "A sustained type I IFN-neutrophil-IL-18 axis drives pathology during mucosal viral infection" for consideration by *eLife*. Your article has been reviewed by 3 peer reviewers, and the evaluation has been overseen by a Reviewing Editor and Tadatsugu Taniguchi as the Senior Editor. The following individuals involved in review of your submission have agreed to reveal their identity: Jennifer M Lund (Reviewer #1); Amariliz Rivera (Reviewer #2).

We all felt that this was an interesting and important paper but agreed that several additions could strengthen the work.

Essential revisions:

1. Establish IFN at the protein level on day 5 and 7 to confirm continued production.

2. List all neutrophil derived ISGs in a table

3. Document neutrophil depletion in the vaginal mucosa over the first 5 days following 1A8 injection.

4. Please compare HSV viral loads in neutrophil depleted mice until clearance or death versus non-neutrophil depleted mice. Consider measurement of viral load in CNS as well. Please also compare HSV-1 and HSV-2 vaginal viral loads to assess for differential priming of neutrophil responses.

The reasons for these requests are outlined in more detail below.

*Reviewer #1 (Recommendations for the authors):*

• Figure 1A – Please show gating strategy for flow cytometry data in Figure 1, both part A and Supplemental.

• In lines 188-190, the authors state that their data "suggests that the regulation of the neutrophil response after HSV-1 or HSV-2 infection was distinct, leading to disparate inflammatory outcomes". However, they have not proven cause and effect here, as there is more than just the neutrophil response that is different between these two infections. This sentence should be revised to suggest an association rather than cause and effect.

*Reviewer #2 (Recommendations for the authors):*

1. The rationale for examining neutrophil-specific STIM1/STIM2 mice should be better articulated. Sentence 166 seems incomplete.

2. Sentence 431 is also unclear/incomplete3 The co-localization of S100A8 and IL-18 staining shown in Figure 6F is hard to appreciate. It would be helpful if the authors show higher magnification images.

3. The levels of IL-18 measured by ELISA in Figure 6A-6C seems very small relative to abundant staining seen by immunofluorescence in Figure 6F. The addition of control staining for naïve mice would be helpful as reference. The authors might want to also add to the discussion the possibility that the immunofluorescence could be detecting inactive, intracellular IL-18 while ELISA would (presumably) be measuring secreted IL-18.

4. To solidify a direct link from type I IFNs on neutrophils to IL-18 the authors might want to consider ex vivo culture of neutrophils and type I IFN stimulation followed by analysis if IL-18 release in the culture supernatant. I understand that the mechanisms are more complex than a simple model where all the IL-18 is coming directly from neutrophils but it would have been useful to check and if the authors checked and the answer was negative it would have also been informative to the let the readers know.

*Reviewer #3 (Recommendations for the authors):*

– In Figure 2B there are some mice where as many neutrophils were present in the tissue at day 0 as there were after infection. Why is this?

– The difference between the two infections, HSV-2 and HSV-1, in terms of disease and the role of neutrophils is interesting. The data suggest that in both infections neutrophils are recruited to the tissue and respond by upregulating ISGs. What is lacking is mechanistic understanding of why neutrophils drive disease during HSV-2 infection whereas during HSV-1 infection neutrophils are present yet do not drive inflammation. Is the main cause of the difference in disease outcomes the persistence or higher titers of virus during HSV-2 infection that results in sustained stimulation of the neutrophils? Or are viral titers the same between the two infections? This is important because it speaks to the mechanism of sustained IFN-I production (Figure 3E) and sustained signaling to neutrophils and a bit more discussion of this would be helpful.

– The data on HSV-1 shown in Figure 2 supplement 1 are important and would be better placed in the main figure. The data seem to suggest that disease is actually worse in in HSV-1 mice where neutrophils were depleted. Viral titer data should also be shown from this experiment to demonstrate any change.

– The data in Figure 2C require quantification of the images to support the conclusion being made about interactions.

– IL-18-floxed mice (10.1016/j.cell.2015.10.072) would be a good way to define the role of IL-18 from neutrophils versus other cells types. Have the authors considered this approach?

---

## [Author Response]

Essential revisions:1. Establish IFN at the protein level on day 5 and 7 to confirm continued production.

We have now provided IFNb protein measurement from vaginal washes (secreted IFN) at 4, 5 d.p.i. and 7 d.p.i. as requested, as well as from tissue homogenates at 7 d.p.i., as suggested below in new Figure 3 and new Figure 3 – Supplement 3.

2. List all neutrophil derived ISGs in a table.

Complete lists of differentially expressed genes in neutrophils between HSV-1 and HSV-2 infection at 5 d.p.i, including all ISGs, has been included as Source Files for Figure 3 – Supplement 1. We have also provided tables in new Figure 3 – Supplement 1 indicating the top 20 DEGs in neutrophils between HSV-1 and HSV-2 infection at 5 d.p.i.. As the Hallmark IFNa gene set (publicly available at GSEA) was used to determine ISG expression level, ISGs listed in this gene set have been highlighted in the new figure.

3. Document neutrophil depletion in the vaginal mucosa over the first 5 days following 1A8 injection.

This data (depletion up to 6 d.p.i.) has now been provided in new Figure 1A.

4. Please compare HSV viral loads in neutrophil depleted mice until clearance or death versus non-neutrophil depleted mice. Consider measurement of viral load in CNS as well. Please also compare HSV-1 and HSV-2 vaginal viral loads to assess for differential priming of neutrophil responses.

We have now provided viral load in vaginal washes in the neutrophil depletion experiments and vaginal tissue homogenates of control or neutrophil-depleted mice out to 7 d.p.i. in new Figure 1. As it is unclear when the mice clear WT HSV-2, and to avoid survival bias in the data (please see Author response image 1) as well as to match the inflammation scores, we have chosen 7 d.p.i. as our endpoint. Viral loads in the nervous system are also provided – we have not included them in the actual manuscript in order to maintain focus on genital inflammation. We have also now provided viral titer data in vaginal washes from HSV-1 and HSV-2 infected mice out to 6 d.p.i. (data adapted from our previously published study) and have also measured viral burden in the vaginal tissue at 7 d.p.i. in new Figure 2 – Supplement 1.

**Author response image 1. sa2fig1:** Mice depleted of neutrophils have a higher survival rate compared to controls. Survival was monitored over 14 days after infection in isotype control or anti-Ly6G mAb-treated mice. Curves are significantly different by log-rank test (p<0.0001).

We had originally anticipate difficulties in providing 7 d.p.i. data from vaginal washes; however, as we were able to overcome this challenge by modifying our collection technique. We thank the reviewers for being flexible on this matter, and have now provided data from both washes and tissue homogenates, as suggested below.

The reasons for these requests are outlined in more detail below.Reviewer #1 (Recommendations for the authors):• Figure 1A – Please show gating strategy for flow cytometry data in Figure 1, both part A and Supplemental.

This has now been provided in new Figure 1A and new Figure 1 – Supplement 2A.

• In lines 188-190, the authors state that their data "suggests that the regulation of the neutrophil response after HSV-1 or HSV-2 infection was distinct, leading to disparate inflammatory outcomes". However, they have not proven cause and effect here, as there is more than just the neutrophil response that is different between these two infections. This sentence should be revised to suggest an association rather than cause and effect.

We apologize for the incorrect wording. This sentence (new lines 217-218) has now been revised.

Reviewer #2 (Recommendations for the authors):1. The rationale for examining neutrophil-specific STIM1/STIM2 mice should be better articulated. Sentence 166 seems incomplete.

Apologies for this error – the sentence has been fixed and a more thorough rationale for the use of STIM1/STIM2-deficient mice has been provided in lines 184-187 of the revised manuscript.

2. Sentence 431 is also unclear/incomplete3 The co-localization of S100A8 and IL-18 staining shown in Figure 6F is hard to appreciate. It would be helpful if the authors show higher magnification images.

Clearer images have now been included as insets in new Figure 6F.

3. The levels of IL-18 measured by ELISA in Figure 6A-6C seems very small relative to abundant staining seen by immunofluorescence in Figure 6F. The addition of control staining for naïve mice would be helpful as reference. The authors might want to also add to the discussion the possibility that the immunofluorescence could be detecting inactive, intracellular IL-18 while ELISA would (presumably) be measuring secreted IL-18.

We agree that the staining patterns suggest that the IF may be detecting more than just bioactive IL-18. There is unfortunately not enough information provided by the manufacturer to determine whether the antibody used for IF is detecting only bioactive IL-18 or all forms of IL-18. We have added this caveat to the revised manuscript at lines 481-482. We have also included the naive mouse control of our IL-18 IF staining in new Figure 6F.

4. To solidify a direct link from type I IFNs on neutrophils to IL-18 the authors might want to consider ex vivo culture of neutrophils and type I IFN stimulation followed by analysis if IL-18 release in the culture supernatant. I understand that the mechanisms are more complex than a simple model where all the IL-18 is coming directly from neutrophils but it would have been useful to check and if the authors checked and the answer was negative it would have also been informative to the let the readers know.

This is an excellent suggestion and would be very helpful in determining whether neutrophils are indeed the source of IL-18 driving pathology during HSV-2 infection. This experiment will be included in future studies in which the mechanism by which type I IFN regulates excess IL-18 levels in the vagina are elucidated.

Reviewer #3 (Recommendations for the authors):– In Figure 2B there are some mice where as many neutrophils were present in the tissue at day 0 as there were after infection. Why is this?

The spread in neutrophil numbers is something that we often see in our mice, and unfortunately we do not have any concrete answers to offer in explanation. We can only speculate that there may be some differences in neutrophil infiltration depending on the time of day that the cells were collected, or perhaps the length of time the mice have been housed in our animal facility. There may also be some variability stemming from the collection of the samples. We can, however, offer assurance that all of the mice used in this study were from the same vendor or bred in a single room in our animal facility, similar in age and housed in the same room to minimize any variability in genetic background, microbiome and other environmental factors that may have affected host responses.

– The difference between the two infections, HSV-2 and HSV-1, in terms of disease and the role of neutrophils is interesting. The data suggest that in both infections neutrophils are recruited to the tissue and respond by upregulating ISGs. What is lacking is mechanistic understanding of why neutrophils drive disease during HSV-2 infection whereas during HSV-1 infection neutrophils are present yet do not drive inflammation. Is the main cause of the difference in disease outcomes the persistence or higher titers of virus during HSV-2 infection that results in sustained stimulation of the neutrophils? Or are viral titers the same between the two infections? This is important because it speaks to the mechanism of sustained IFN-I production (Figure 3E) and sustained signaling to neutrophils and a bit more discussion of this would be helpful.

This is an excellent question and is the subject of ongoing study in our lab. Although we do not yet fully understand what drives sustained type I IFN production and stimulation of neutrophils, we can speculate that it is unlikely due to differences in replicating virus load between the two infection models, as differences in IFNb production do not correlate with the minor differences in viral clearance between HSV-1 and HSV-2 (see new Figure 3 and Figure 2 – Supplement 1). However, it is possible that there may be other virological differences that drive severe inflammatory disease during HSV-2 but not during HSV-1, including production of defective particles or differences in viral dissemination to the nervous system (Lee et al., JCI Insight 2020). It is also possible that early disparities in the inflammatory environment after HSV-1 or HSV-2 infection are modulating neutrophil responses to type I IFN signaling later during infection. We have included discussion on this point at lines 408-419 in the revised manuscript.

– The data on HSV-1 shown in Figure 2 supplement 1 are important and would be better placed in the main figure. The data seem to suggest that disease is actually worse in in HSV-1 mice where neutrophils were depleted. Viral titer data should also be shown from this experiment to demonstrate any change.

We have now included the inflammation score data from previous Figure 2 – Supplement 1 in new Figure 2, along with the corresponding virus quantification in vaginal washes and tissue as new Figure 2 – Supplement 1C. Similar to HSV-2, we did not detect marked changes in viral burden after HSV-1 infection in the absence of neutrophils.

– The data in Figure 2C require quantification of the images to support the conclusion being made about interactions.

Agreed – this sentence has been removed.

– IL-18-floxed mice (10.1016/j.cell.2015.10.072) would be a good way to define the role of IL-18 from neutrophils versus other cells types. Have the authors considered this approach?

We completely agree that this would be an excellent approach, and are currently in the process of obtaining these mice.